# Spatiotemporal Evolution of the Land Cover over Deception Island, Antarctica, Its Driving Mechanisms, and Its Impact on the Shortwave Albedo

Javier F. Calleja [1,*], Rubén Muñiz [2], Jaime Otero [3], Francisco Navarro [3], Alejandro Corbea-Pérez [4], Carleen Reijmer [5], Miguel Ángel de Pablo [6] and Susana Fernández [7]

1 Departamento de Física, Universidad de Oviedo, 33007 Oviedo, Spain
2 Departamento de Informática, Universidad de Oviedo, 33203 Gijón, Spain; rubenms@uniovi.es
3 Departamento de Matemática Aplicada a las TIC, Universidad Politécnica de Madrid, 28040 Madrid, Spain; jaime.otero@upm.es (J.O.); francisco.navarro@upm.es (F.N.)
4 Departamento de Ingeniería Informática y de Sistemas, Universidad de La Laguna, 38200 San Cristóbal de La Laguna, Spain; alejandrocorbea@gmail.com
5 Physics Department, Utrecht University, 3584 Utrecht, The Netherlands; c.h.tijm-reijmer@uu.nl
6 Departamento de Geología, Geografía y Medio Ambiente, Universidad de Alcalá, 28805 Alcalá de Henares, Spain; miguelangel.depablo@uah.es
7 Departamento de Geología, Universidad de Oviedo, 33005 Oviedo, Spain; fernandezmsusana@uniovi.es
* Correspondence: jfcalleja@uniovi.es

**Abstract:** The aim of this work is to provide a full description of how air temperature and solar radiation induce changes in the land cover over an Antarctic site. We use shortwave broadband albedo (albedo integrated in the range 300–3000 nm) from a spaceborne sensor and from field surveys to calculate the monthly relative abundance of landscape units. Field albedo data were collected in January 2019 using a portable albedometer over seven landscape units: clean fresh snow; clean old snow; rugged landscape composed of dirty snow with disperse pyroclasts and rocky outcrops; dirty snow; stripes of bare soil and snow; shallow snow with small bare soil patches; and bare soil. The MODIS MCD43A3 daily albedo products were downloaded using the Google Earth Engine API from the 2000–2001 season to the 2020–2021 season. Each landscape unit was characterized by an albedo normal distribution. The monthly relative abundances of the landscape units were calculated by fitting a linear combination of the normal distributions to a histogram of the MODIS monthly mean albedo. The monthly relative abundance of the landscape unit consisting of rugged landscape composed of dirty snow with dispersed clasts and small rocky outcrops exhibits a high positive linear correlation with the monthly mean albedo ($R^2 = 0.87$) and a high negative linear correlation with the monthly mean air temperature ($R^2 = 0.69$). The increase in the solar radiation energy flux from September to December coincides with the decrease in the relative abundance of the landscape unit composed of dirty snow with dispersed clasts and small rocky outcrops. We propose a mechanism to describe the evolution of the landscape: uncovered pyroclasts act as melting centers favoring the melting of surrounding snow. Ash does not play a decisive role in the melting of the snow. The results also explain the observed decrease in the thaw depth of the permafrost on the island in the period 2006–2014, resulting from an increase in the snow cover over the whole island.

**Keywords:** cryosphere; albedo; landscape; air temperature; solar radiation

## 1. Introduction

The cover of snow and ice in Antarctica plays a crucial role in the atmosphere and surface energy budget. At a global scale, it explains the different roles played by the shortwave radiation in the Arctic and the Antarctic [1]. At a local scale, at locations with exposed bare soil, the relative amount of snow-covered and snow-free surfaces determines the surface energy balance [2,3]. The impact of albedo distribution on melt rates has been

quantified. The shifts in the dominant surface type from snow to bare ice and clean ice to impurity-rich surfaces are important drivers in increasing seasonal ice sheet melt rates [4]. Moreover, in places with permafrost, the spatial extent and thickness of the snow cover play a crucial role in the freezing and thawing of the permafrost active layer [5]. The main factor controlling the thaw depth is the duration and thickness of the snow cover, which, above certain thresholds, insulate the ground from warming [6]. A detailed description of the evolution of the land cover at sites with varying extensions of snow-covered and snow-free surfaces is mandatory to gain insight into the processes mentioned above. Deception Island, in the South Shetland Islands, is a site with a mixture of bare soil and snow, with the amount of surface occupied by each varying along the season, making it a suitable place to carry out these kinds of studies; and, as an extra motivation for this work, Deception Island is also a site with widespread permafrost [7].

Information on the relative abundance of snow or bare soil can be obtained from albedo data. The albedo of the surface in areas covered totally or partially by snow and ice depends on the properties of the snow and ice and on the relative amount of snow, ice, and bare soil. Snow albedo evolves over time at different time scales: it can change abruptly in few hours, and it also exhibits daily, monthly, and seasonal variations, along with long-term variations (trend) [8]. Moreover, in areas with a shallow snow cover and where snowmelt is ubiquitous, the albedo of the surface varies over the summer with the exposition of bare soil. Although high-resolution satellite sensors may provide an accurate, instantaneous picture of the land cover, their low temporal resolution is a serious drawback to obtain long time series with sufficient temporal resolution. When using mid-resolution satellite sensors, the sole estimation of the mean albedo in such cases provides a poor description of the real picture. For example, a decrease in albedo can be due to the metamorphization of the snow or an increase in the abundance of bare soil. On the other hand, albedo can increase if new, fresh snow falls on old snow, like in a snowfall event, or if the area covered by snow increases.

The aim of this work is to provide a precise description of the land cover evolution and its driving mechanisms over Deception Island, Antarctica. In order to do so, we will show that the combined use of satellite and field shortwave broadband albedo measurements (albedo integrated in the range 300–3000 nm) can provide an accurate description of the land cover over an Antarctic site. The advantage of the method presented lies in the fact that field measurements are easy to carry out, even in the harsh environment of Deception Island, since the equipment is rough, light, and easy to transport over snowed and iced areas.

This work is organized as follows. In Section 2, we provide a description of the study area, the data used, and the data processing. In Section 3, we present the results. In Section 4, we discuss the results obtained, presenting the mechanism proposed to explain the evolution of the landscape over Deception Island and its driving mechanisms.

## 2. Materials and Methods

In this section, we describe the materials (study area, in situ and field data, and satellite data) and the methodology (data processing).

### 2.1. Study Area

The study area is Deception Island, located in the South Shetland Islands Archipelago on the NW coast of the Antarctic Peninsula (Figure 1). Deception Island is an active volcano with three recent eruptions in 1967, 1969, and 1970 [9] and unrest episodes in 1992, 1997, and 2015 [10,11]. As a result of the recent eruptions, Deception Island is covered by volcanic ash and pyroclasts, and many of the glaciers remain ash-covered today. The soils are composed of ashes and pyroclasts with high porosity [6]. It has areas of dark soil of varying extension during summer as well as large areas of snow covered by volcanic ash and lapilli. The changing mixture of bare soil and snow/ice makes it a suitable location for detecting changes in the land cover. The Spanish Antarctic Station Gabriel de Castilla is located on the island, providing access and long-term series of meteorological observations. The

climate is cold oceanic, with frequent summer rainfall, a moderate annual temperature range (10 °C monthly mean deviations), and mean annual air temperatures close to −3 °C at sea level [12]. A CALM (Circumpolar Active Layer Monitoring) site (The Crater Lake CALM-S) is located on the island to monitor the status of the permafrost active layer; it is placed in a small and relatively flat plateau-like step covered by volcanic and pyroclastic sediments at 85 m a.s.l. (62°59′06.7″S, 60°40′44.8″W).

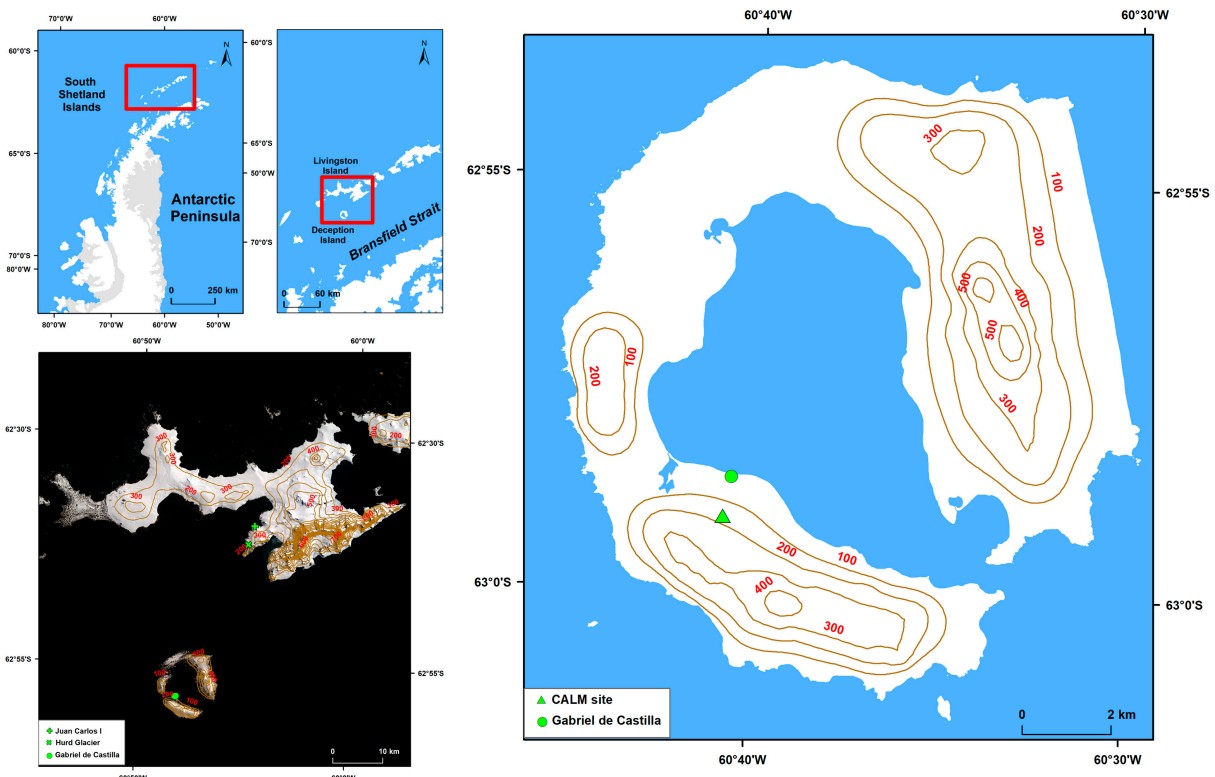

**Figure 1.** Location of Deception Island and Livingston Island, including the location of the AWSs and the CALM site.

### 2.2. In Situ and Field Data

We have used data from three Automatic Weather Stations (AWSs) and data collected over different locations on Deception Island with a portable albedometer. The data collected with the portable albedometer will be referred to as field data from now on.

Two AWSs are located on Livingston Island (Figure 1, Table 1): one close to Juan Carlos I Station (JCI AWS); one located on Hurd Glacier (HG AWS); and the third AWS is in the vicinity of Gabriel de Castilla Antarctic Station on Deception Island (GdC AWS) (Figure 1, Table 1). From JCI and HG AWSs, we used radiation data of 25 January 2019, 31 January 2019, and 1 February 2019 for the sensitivity analysis of field albedo measurements, as explained in Appendix B. The sensitivity analysis consists of comparing the field data to data collected by an AWS (taken as truth). This allowed us to evaluate the impact of several factors on the field data: the response of the sensors to changing incident irradiance, the body of the researcher carrying the sensors, and clouds. The sensitivity analysis could not be carried out on Deception Island because the GdC AWS does not provide reflected shortwave radiation data.

GdC AWS has been providing air temperature and incident sun radiation data every 10 min since 1/09/2005 [13]. The air temperature and the radiation data from GdC AWS are going to be used to study the evolution of albedo and landscape units as a function of radiation and temperature. All the data from JCI and GdC AWSs can be downloaded from the AEMET (Spanish Meteorological Service) webpage [13]. Albedo data from AWSs are only needed for sensitivity analysis purposes. For the investigation of the driving

mechanism of the evolution of the landscape units, we need incident radiation and air temperature measurements as close as possible to the study area. These data are available at GdC AWS on Deception Island.

**Table 1.** Location of the Automatic Weather Stations and instrumentation used. The last row describes the purpose of the data in this study. The values in brackets are the uncertainties.

| Livingston Island | Deception Island |
| --- | --- |
| Juan Carlos I (JCI)<br>62°39′48″S 60°23′19″W; ASL: 13 m<br>Hurd Glacier (HG)<br>62°41′48″S 60°24′44″W; ASL: 140 m | Gabriel de Castilla (GdC)<br>62°58′38″S 60°40′31″W; ASL: 13 m. Since 4 February 2005<br>moved to<br>62°58′38″S 60°40′33″W; ASL: 12 m. Since 12 February 2007 |
| Albedo KIP-ZONEN CM11 (<±1%) | Incident solar radiation KIP-ZONEN CM11 ($\pm 10$ W/m$^2$)<br>Air temperature HMP45C ($\pm 0.4$ °C) |
| Sensitivity analysis of distributed albedo measurements<br>Ratio of diffuse to global radiation to discuss the<br>cloudiness in the area | Solar radiation and air temperature over the study site as driving<br>mechanisms of albedo and landscape evolution |

Some variables exhibit a great dependence on topography. Wind velocity, wind direction, and air temperature have been measured at AWSs on Deception Island and on King George Island, located 120 km apart [14]. The results show that wind direction and velocity on Deception Island follow a very different behavior to that measured at King George Island, while daily air temperature at Deception Island follows the same temporal trend as daily air temperatures measured at King George Island. According to these results, we think that we can take the air temperature measured at the GdC AWS as representative of the whole island.

We also collected field albedo data in January and February 2019 over different locations on Deception Island. Field albedo measurements were carried out using a homemade portable albedometer consisting of two pyranometers, one facing the sky and another facing the surface, and two synchronized dataloggers (Figure 2). We used Class C DeltaOHM LPPYRA03 pyranometers, based on the thermopile principle, with a viewing solid angle of $2\pi$ sr, a spectral range of $300-2800$ nm, and HD2102.2 DeltaOHM dataloggers. The pyranometers are rigidly attached to a pole which is, in turn, rigidly attached to a back rack carried by the researcher. The ratio of the signal from the pyranometer facing the surface to the signal of the pyranometer facing the sky provides the albedo of the surface. The sampling frequency was 0.2 Hz. The selected sampling sites were flat, and the pyranometers were kept horizontal by visual inspection using a bubble level. Measurements were carried out on different locations inside each sampling site to obtain the albedo variation due to surface changes. Data were collected while standing still with the portable albedometer carried as shown in Figure 2 at around local noon. Seven sampling sites were chosen corresponding to seven landscape units representative of the landscape of Deception Island. The selection of the landscape units was performed following the expert criteria, taking into account the definition of a landscape unit as a homogeneous tract of land at the scale at issue [15]. In this work, we use data from a spaceborne sensor with a spatial resolution of 500 m. The chosen landscape units spread over areas with a characteristic length of at least that size. The identification codes of the landscape units, the location of the sampling sites, the data and time of the data acquisition, and the sky conditions during the field data collection are shown in Table 2. The identification code of each landscape unit is the mean value of the broadband albedo; this makes the reading easier. The landscape units are (Figure 3, Table 3) continuous clean fresh snow (0830); continuous clean old snow (0736); rugged landscape composed of dirty snow with dispersed pyroclasts and rocky outcrops (0599); continuous dirty snow (0457); stripped mixture of snow and bare soil (0313); shallow snow with small bare soil patches (0166); and continuous bare soil (0041). Although both 0313 and 0166 are a mixture of bare soil and shallow snow, they differ in the distribution

pattern of snow and bare soil, and they coexist throughout the season. Clean fresh snow is barely seen over Deception Island in the months of January and February, when the data were collected. We took advantage of a snowfall event on 18 February 2019 to sample some clean fresh snow that disappeared very quickly. We assume that this kind of land cover can be present at the beginning of the season, so we included it as a landscape unit. Landscape unit 0166 appears as a consequence of light snow events covering bare soil areas. The footprint of the field data is estimated to be a circle with a diameter ten times the height of the sensors [16]. In our case, the sensors are 1.5 m above the ground, so the footprint of the field data is a circle of a diameter of 15 m. For each landscape unit, we checked that the samples were distributed according to a normal distribution. This is especially critical in the case of landscape unit 0313: if the sensors were placed too low, we would sample bare soil and snow separately, yielding a binomial distribution.

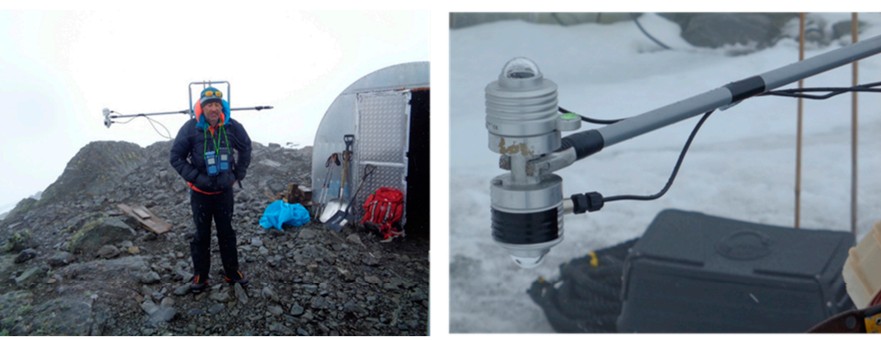

**Figure 2.** Portable albedometer and experimental set-up.

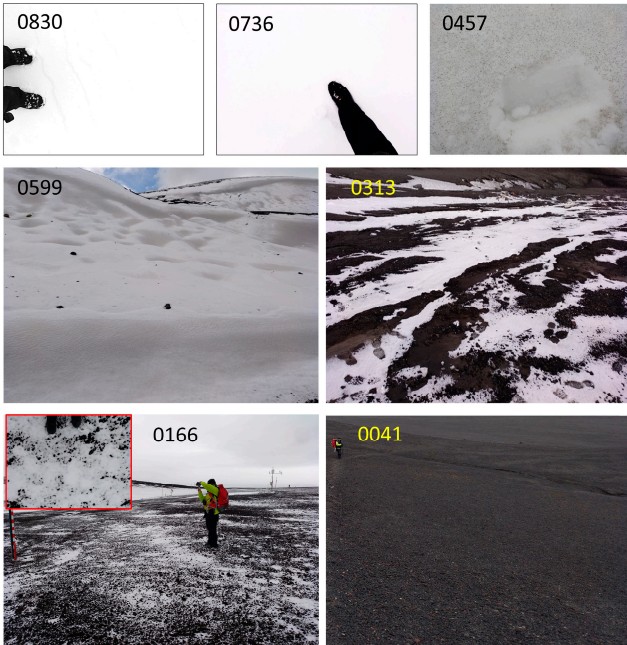

**Figure 3.** Landscape units and their identification codes; see also Table 3. The inset in 0166 corresponds to a nadir view of the cover. The large area covered by 0166 in the lower left panel is the CALM site.

**Table 2.** Location of the sampling sites, sampling date, time of data acquisition, and sky conditions during sampling for each of the landscape units. LST = Local Solar Time.

| L | Location | Date | Time | Sky |
|---|---|---|---|---|
| 0830 | 62°58′41.7″S; 60°40′42.7″W | 18 January 2019 | 12:44–15:10 | Overcast |
| 0736 | 62°58′52.0″S; 60°40′55.6″W | 14 January 2019 | 10:32–10:46 | Overcast |
| 0599 | 62°58′46.0″S; 60°40′41.6″W | 11 January 2029 | 10:56–11:26 | Partly cloudy |
| 0457 | 62°59′10.9″S; 60°40′39.2″W | 09 January 2019 | 11:05–12:09 | Overcast |
| 0313 | 62°58′39.0″S; 60°40′38.3″W | 11 January 2019 | 9:00–9:30 | Partly cloudy |
| 0166 | 62°59′08.1″S; 60°40′43.7″W | 08 January 2019 | 12:15–12:30 | Overcast |
| 0041 | 62°59′07.8″S; 60°40′42.8″W | 08 January 2019 | 12:20–14:00 | Overcast |

**Table 3.** Landscape unit code ($L$), mean albedo ($\mu_L$), and standard deviation ($\sigma_L$) of the normal distribution of each landscape unit.

| L | Description | $\mu_L$ | $\sigma_L$ |
|---|---|---|---|
| 0830 | Clean fresh snow | 0.830 | 0.016 |
| 0736 | Clean old snow | 0.736 | 0.013 |
| 0599 | Rugged landscape of snow and pyroclasts | 0.599 | 0.040 |
| 0457 | Dirty snow | 0.457 | 0.018 |
| 0313 | Stripes of bare soil and snow | 0.313 | 0.080 |
| 0166 | Shallow snow and bare soil holes | 0.166 | 0.053 |
| 0041 | Bare soil | 0.041 | 0.009 |

*2.3. Satellite Data*

MODIS daily albedo product MCD43A3 (C6) was used in this work [17]. The time span was from the 2000–2001 season to the 2020–2021 season. Only data with Sun Zenith Angle (SZA) < $75^0$ are considered; this means that a season spans from September 1 of a year to March 31 of the next year. Data were downloaded using the Google Earth Engine API [18]. MCD43A3 includes one band of shortwave Black Sky Albedo (BSA) and one band of shortwave White Sky Albedo (WSA). In this work, we present the results obtained using the shortwave BSA band. In the calculation of the albedo, the incident radiance can be divided into a direct component with angles θ, φ ($E_{dir}(\theta,\varphi)$), and a diffuse component $L_{diff}$, and assuming that $L_{diff}$ is isotropic, we define the fraction of diffuse radiation as d = ($\pi L_{diff}/E_{dir}$) [19]. The actual albedo of the surface, the so-called blue-sky albedo, can be obtained from WSA and BSA as:

$$\alpha = d \times WSA + (1 - d) \times BSA \tag{1}$$

where the calculation of the actual blue-sky albedo $\alpha$ requires the fraction of diffuse radiation ($d$ in Equation (1)). The maximum relative difference between WSA and BSA in this work is 3%. The relative difference between $\alpha$ and BSA and $\alpha$ and WSA is below that value [20]. So, we take BSA as the blue-sky albedo in this work.

*2.4. Methods*

In this section, we describe the processing of in situ and field data to obtain the albedo normal distribution of each landscape unit, the satellite data processing to calculate the monthly average albedo and the corresponding monthly histogram, and the fit of the monthly average histogram to the normal distributions of the landscape units.

2.4.1. In Situ and Field Data Processing

Regarding the data from the GdC AWS, daily means were calculated only if data were available for at least 80% of the 10-minute daily data, and monthly and seasonal means of meteorological variables were calculated only if daily means were available for at least 80% of the days [21].

Regarding field albedo measurements with the portable albedometer, we have to be sure that the variations in the albedo inside a sampling site are exclusively due to variations in the surface. While soil albedo is known to be insensitive to changes in incident irradiance [22], snow albedo depends on cloudiness to a great extent [23,24]. Furthermore, because we intend to describe each landscape unit by a normal distribution, we must be able to establish the minimum albedo variation that can be detected with the portable albedometer in order to ensure that the standard deviation of the normal distribution is above the minimum albedo variation that can be measured. Because of time constraints and logistics during the Antarctic campaign, some of the albedo measurements had to be performed under cloudy conditions. Because these data are going to be compared with satellite data, which are supposed to be cloud-free, field albedo data must be corrected for the effect of clouds. In general, field albedo measurements are affected by the following factors:

1.  Fluctuations due to incident irradiance oscillations (changes in the incident irradiance during the sample collection due to varying atmospheric conditions or to uncontrolled tilt of the pyranometers). Data must be corrected for this effect to ensure that albedo variations inside a sampling site are exclusively due to changes in the surface; the influence of fluctuations of incident irradiance on the albedo will also allow us to estimate the precision of the portable albedometer;

2.  Bias caused by the experimental set-up (pyranometers model, influence of the body of the researcher carrying the pyranometers);

3.  Cloudiness (field data are going to be compared with satellite data, which are supposed to be cloud-free data).

A detailed description of these points is provided in Appendix B. In this work, field albedo data were only corrected for fluctuations in the incident irradiance. Bias and cloudiness are introduced as uncertainties in the albedo, and their impact is discussed in Appendix B. This is because a correction of bias and cloudiness for each cover type is impossible, but we can still estimate uncertainties. In Appendix B, we demonstrate that although bias and clouds have an impact on the values of the albedo, the general trends and the discussion of the results of this work are not affected by them. Taking into account these considerations, after collecting the samples, we proceeded as follows: (1) Estimation of albedo fluctuations due to changes in incident irradiance; (2) data filtering and calculation of histograms and fit to normal distributions. Datasets from each sampling site were filtered, eliminating outliers in incident and reflected irradiance until building a set with a coefficient of variation of incident irradiance below 6% to ensure that the variations in albedo were exclusively due to changes in the surface (Appendix B). Then, we checked that there was no correlation between the incident irradiance and the albedo. With the remaining dataset, we built a histogram for each landscape unit $L$ ($L$ = 0041, 0166, 0313, 0457, 0599, 0736, 0830). Each histogram was fitted to a normal distribution $N(\mu_L, \sigma_L)$. The results are shown in Table 3 and Figure 4.

### 2.4.2. Satellite Data Processing

We have calculated monthly averages from September to March. We calculated the monthly average of a pixel only if two conditions were met: (1) There were at least 4 albedo data points in the month; (2) The time interval between the first and the last data point was larger than 15 days. These conditions were established to ensure that the mean average tracked the evolution of the snow cover; we had at least 4 albedo data points, and they were evenly distributed over the month. The resulting monthly average image has several gaps because many pixels did not meet the conditions above. Several gap-filling algorithms have been proposed, but we did not use them to avoid artifacts in the data. We instead devised a procedure to label a given monthly image as representative of the island. We divided Deception Island in 16 sectors of the same size and calculated, for each monthly average image, the number of pixels with data in each sector. A sector was considered representative if the percentage of pixels with data was above 20%. Then, for an image to be taken as representative of the whole island, at least 10 sectors (out of 16) must have

been labeled as representative. We are aware that this procedure reduces the number of months available for the analysis, but on the other hand, we assure that no artifacts were introduced due to excessive data processing. Once a monthly average albedo image was labeled as representative of Deception Island, a histogram of the frequency of albedo was built. We also calculated, for each representative image, the mean monthly albedo over Deception Island, which will be called <α> from now on.

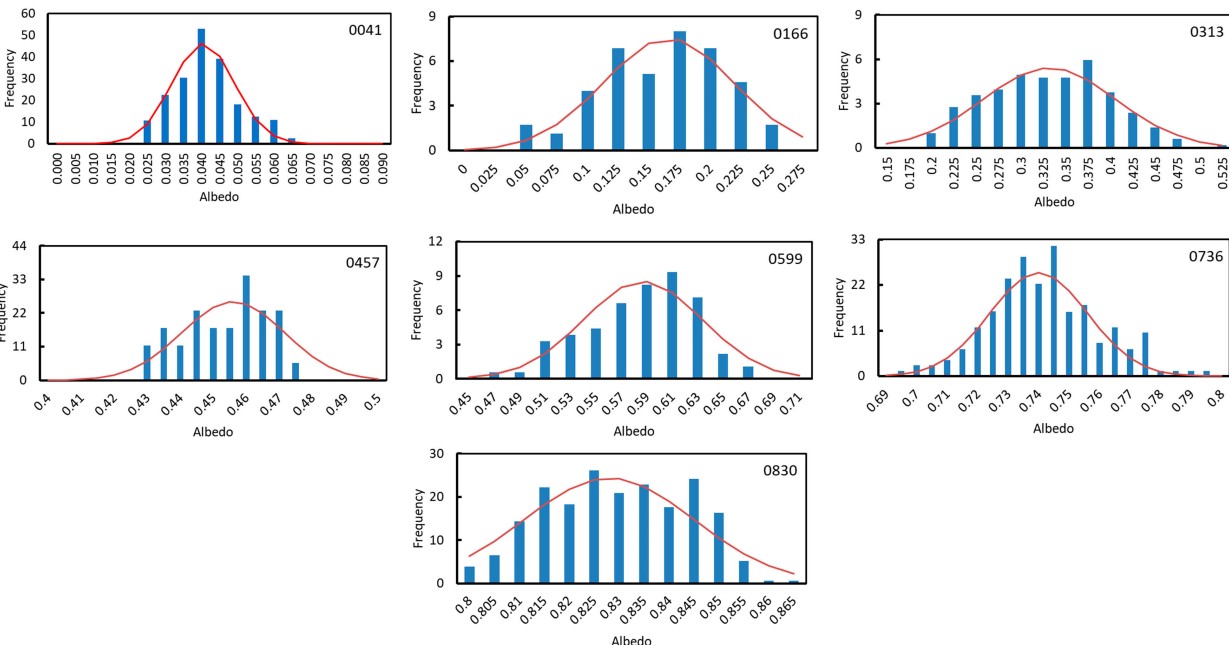

**Figure 4.** Broadband albedo histograms obtained from albedo distributed measurements over the sampling plots of each landscape unit. The solid red line is the normal distribution that best fits to the histogram. Both the histogram and the normal distribution are normalized to the unit area. Note the different scale in the Y-axis.

### 2.4.3. Satellite Albedo Histogram Fit to Field Albedo Normal Distributions

To estimate the contribution of each landscape unit, characterized by a normal distribution, to the satellite data, a linear combination of the probability density function of each normal distribution is performed.

Let $M$ be a month and $S$ a season ($M$ = S, O, N, D, J, F, M; $S$ = 2000–2001, 2001–2002, …, 2020–2021), and let $L$ denote a landscape unit ($L$ = 0041, 0166, 0313, 0457, 0599, 0736, 0830). The probability density function of each landscape unit $L$ was calculated as explained in Section 2.4.1, and a histogram that shows the frequency of pixels of each bin of MODIS monthly mean was constructed following the procedure explained in Section 2.4.2. We denote by $H_{iMS}$ the frequency of bin $i$ for month $M$ and season $S$ in the MODIS histogram.

The area below the normal distribution of landscape unit $L$ in each bin $H_{iMS}$ is calculated from the corresponding probability density function.

$$F_L(x_{iMS}) - F_L\left(x_{(i-1)MS}\right) = \frac{1}{\sigma_L\sqrt{2\pi}}\int_{x_{(i-1)MS}}^{x_{iMS}} e^{\frac{-(u-\mu_L)^2}{2\sigma_L^2}}\,du,$$

$$L = 0041, 0166, 0313, 0457, 0599, 0736, 0830;\ i = 1,\dots,N$$

(2)

with $F_L(x)$ as the probability density function of landscape unit $L$..$N$ is the number of bins; $\mu_L$ and $\sigma_L$ are the mean albedo and the standard deviation of the normal distribution of landscape unit $L$, respectively; $u$ denotes the albedo; and $[x_{(i-1)MS}, x_{iMS}]$ represents the *ith* bin of the histogram of month $M$ of season $S$.

Then, a linear combination of these areas is performed for each bin and the residuals from the pixels' frequency are obtained.

$$r_{iMS} = H_{iMS} - \sum_L a_{LMS}\left(F_L(x_{iMS}) - F_L\left(x_{(i-1)MS}\right)\right), \ i = 1, \ldots, N \tag{3}$$

The coefficients of the linear combination ($a_{LMS}$) are estimated, minimizing the residuals using the least squares method. These coefficients are interpreted as the relative abundance of landscape unit $L$ in month $M$ of season $S$. Finally, the best fit of a linear combination of the normal distributions to each histogram is obtained. To validate the results of the fitting, the root-mean-square error (RMSE) of the residuals is calculated, obtaining a value of 5%.

We then calculated the relative abundance of each landscape unit for each month (mean monthly relative abundance) in the period 2000–2001 to 2020–2021 as:

$$a_{LM} = \frac{\sum_{S=1}^{N_M} a_{LMS}}{N_M} \tag{4}$$

where $a_{LM}$ is the relative abundance of landscape unit $L$ in month $M$; $NM$ is the number of months $M$ in the study (see Table 4).

**Table 4.** Number of months with a representative monthly albedo image in the period 2000–2001 to 2020–2021 ($N_M$). Number of months with presence of each landscape unit. In parentheses are the percentages of months in which each landscape unit is present.

|  | September | October | November | December | January | February | March |
|---|---|---|---|---|---|---|---|
| $N_M$ | 21 | 18 | 18 | 15 | 9 | 6 | 19 |
| N (0830) | 7 (33%) | 3 (17%) | 0 | 0 | 0 | 0 | 0 |
| N (0736) | 21 (100%) | 18 (100%) | 7 (39%) | 1 (7%) | 0 | 0 | 0 |
| N (0599) | 21 (100%) | 18 (100%) | 18 (100%) | 11 (73%) | 6 (67%) | 2 (33%) | 13 (68%) |
| N (0457) | 21 (100%) | 18 (100%) | 18 (100%) | 15 (100%) | 9 (100%) | 6 (100%) | 19 (100%) |
| N (0313) | 20 (95%) | 18 (100%) | 18 (100%) | 15 (100%) | 9 (100%) | 6 (100%) | 19 (100%) |
| N (0166) | 13 (62%) | 16 (89%) | 18 (100%) | 15 (100%) | 9 (100%) | 6 (100%) | 19 (100%) |
| N (0041) | 2 (10%) | 4 (22%) | 14 (78%) | 14 (100%) | 8 (89%) | 6 (100%) | 19 (100%) |

We also calculated the seasonal relative abundance as:

$$a_{LS} = \frac{\sum_M a_{LMS}}{N_S} \tag{5}$$

where $a_{LS}$ is the abundance of landscape unit $L$ in season $S$; $N_S$ is the number of representative months in season $S$ (for example, 2012–2013 and 2013–2014 seasons are the only seasons with $N_{2012-2013} = N_{2013-2014} = 7$).

## 3. Results

### 3.1. Field Albedo Data

The location of the sampling sites, the data and time of the data acquisition, and the sky conditions during the field data collection are shown in Table 2. The name of the landscape units as well as the mean and the standard deviation of the normal distributions are given in Table 3. Each landscape unit is given the name of the mean albedo. In Figure 4, we show the histograms of the field measurements and the corresponding fits to a normal distribution. The landscape units with the larger standard deviation are those that consist of a mixture of bare soil and snow (0599, 0313, and 0166). On the other hand, those consisting of a single and continuous type of cover (0041, 0457, 0736, and 0830) exhibit a lower standard deviation. This is the expected result. In this work, we deal with the concept of landscape units. Some landscape units are made of a single surface cover (0041, 0457, 0736, and 0830), while others consist of a mixture of surface covers (0166, 0313, and 0599). The standard

deviation of the albedo of the landscape units consisting of a single surface cover is due to variations in that surface cover from sample to sample. The standard deviation of the albedo of the mixed landscape units is due to variations in the surfaces plus variations in the relative abundance of each surface type from sample to sample.

### 3.2. MODIS Data

The number of months meeting the representativeness criteria proposed in Section 2.4.2 over the period 2000–2001 to 2020–2021 is summarized in Table 4. The full list of representative months for each season is given in Table A1 of Appendix A. September is the only month labelled as representative for all the seasons.

Only landscape units 0457 and 0313 are permanently present through the whole season. Clean fresh snow (0830) can only be observed in September and October, and even in those months, its presence is rare. Old snow is ubiquitous in September and October but disappears abruptly from November onward. Landscape unit 0599 is permanently observed from September to November, its presence decaying softly the rest of the season. Bare soil (0041) can barely be observed at the beginning of the season, and it is covered by snowfall events (0166). As the season advances, bare soil becomes one of the dominant landscapes.

The total number of months considered in this study is 106. Only seasons 2012–2013 and 2013–2014 have all the months labelled as representative. In the rest of the seasons, there was at least one month that did not reach the label of representative. The low number of seasons with representative December, January, and February months is due to the high cloudiness in the area in those months. Radiation data have been measured on JCI AWS since 1998 during summer months. The mean values of diffuse and global radiation and their ratio in the period 1998–2014 are shown in Table 5. Although these data correspond to Livingston Island, they can give an idea of the situation over all the South Shetland Islands, including Deception Island.

**Table 5.** Mean daily incident sun energy density ($KJ/m^2$) in November, December, January, and February in the period 1998–2014. Values for global and diffuse radiation are given separately. The ratio of the diffuse to the global radiation is also given. Data from the JCI AWS on Livingston Island.

| Month | November | December | January | February |
|---|---|---|---|---|
| Global | 17,077 | 15,855 | 13,645 | 8678 |
| Diffuse | 9723 | 11,062 | 9599 | 6678 |
| Ratio | 0.57 | 0.70 | 0.70 | 0.77 |

The ratio of diffuse to global radiation is higher in December, January, and February, likely due to an increase in cloudiness in the area, explaining the low number of representative December, January, and February months. However, due to the high number of March months ($N_M = 19$ for March, Table 4), we consider that the resulting month distribution is sufficient to be used to track the albedo evolution over the whole season. Moreover, in Section 3.4, we show that the results obtained for seasons 2012–2013 and 2013–2014 agree with those obtained using the whole set of data, indicating that the missing months do not affect the general discussion.

### 3.3. MODIS Albedo and AWS Air Temperature

In this work, we intend to link albedo to landscape changes. We hypothesize that one of the driving mechanisms of albedo evolution and landscape change is air temperature. It is worth gaining insight in the relationship between albedo and air temperature on Deception Island. The mean monthly albedo ($<\alpha>$) against the mean monthly air temperature from the GdC AWS in the periods from 2006–2007 to 2020–2021 is shown in Figure 5. The monthly mean albedo decreases with temperature at a rate that increases with increasing temperature. The mean monthly albedo attains a maximum value between 0.6 and 0.7.

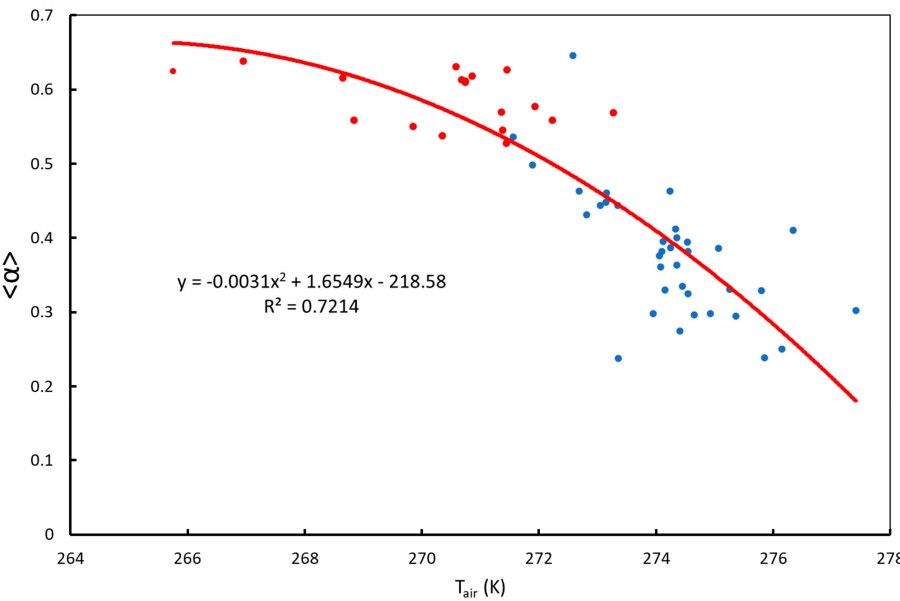

**Figure 5.** Mean monthly albedo over Deception Island versus mean monthly air temperature from GdC AWS in the periods from 2006–2007 to 2020–2021. Red dots correspond to September and October. Blue dots correspond to November, December, January, February and March. The fit is given as a visual guide.

### 3.4. Relative Abundance of Landscape Units

For each landscape unit, we calculated the monthly relative abundance (relative abundance of landscape unit $L$ in month $M$ of season $S$) $a_{LMS}$ as explained in Section 2.4.3. As an example, in Figure 6 we show the albedo histograms and the corresponding fit to a linear combination of $N(\mu_L, \sigma_L)$ for the 2013–2014 season.

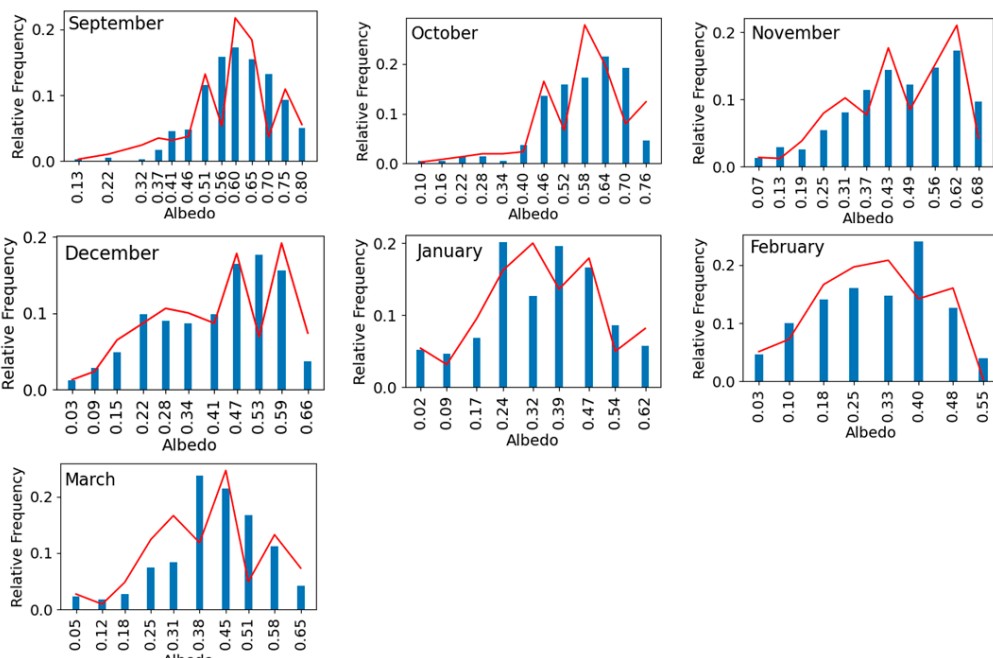

**Figure 6.** Mean monthly albedo histograms (MODIS data, blue bars) for the 2013–2014 season. The red line is the fit to a linear combination of the normal distributions of the landscape units. The Y-axis is the relative frequency, and the X-axis is the albedo. Note that the scale in the ordinates is different in each graph.

### 3.4.1. Mean Monthly Relative Abundance

The mean monthly relative abundances ($a_{LM}$) in the periods from 2000–2001 to 2020–2021 of each landscape unit are shown in Figure 7. Clean snow has a small relative abundance and disappears very soon in the season: 0830 is only present in September and November, and 0736 is present in September, November, and December. On the other hand, bare soil appears in November, and it is present until the end of the season with increasing relative abundance over the season, as expected. The landscape units with the largest relative abundance are 0599 at the beginning of the season and 0313 and 0166 in the second half of the season. It is worth noting that the relative abundance of dirty snow (0457) remains nearly constant over the season, being the only landscape unit with this behavior.

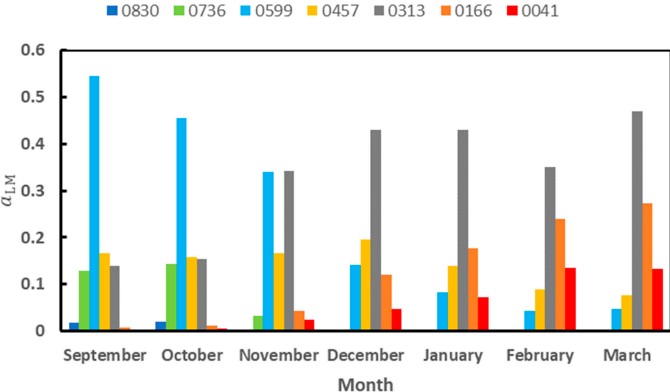

**Figure 7.** Relative abundance of each landscape unit for each month ($a_{LM}$) over the periods from 2000–2001 to 2020–2021.

### 3.4.2. Monthly Relative Abundance and Driving Mechanisms

The relationship between the monthly relative abundance ($a_{LMS}$) and the monthly mean albedo over the whole island and the monthly mean air temperature at the GdC AWS is summarized in Table 6. Landscape unit 0457 (dirty snow) exhibits the lowest correlation with $<\alpha>$ and $T_{air}$ because of its nearly constant presence over the season. Albedo increase seems to be mainly due to the increase in the surface occupied by 0599 and, to a lesser extent, by 0736. The surface occupied by landscape unit 0536 diminishes with increasing $T_{air}$.

**Table 6.** Coefficient of determination between the monthly relative abundance ($a_{LMS}$) and the monthly mean albedo over Deception Island ($<\alpha>$) and the monthly mean air temperature from the GdC AWS ($T_{air}$). Data in red are statistically significant at a 95% level (*p*-value < 0.05). Values with a very high coefficient of determination are marked in bold. The correlation is linear in all the cases, being positive (+) or negative (−) as indicated.

| *L* | 0041 | 0166 | 0313 | 0457 | 0599 | 0736 | 0830 |
|---|---|---|---|---|---|---|---|
| $<\alpha>$ | 0.50 (−) | **0.77 (−)** | **0.63 (−)** | 0.12 | **0.87 (+)** | 0.45 (+) | 0.26 |
| $T_{air}$ | 0.21 (+) | 0.46 (+) | 0.47 (+) | 0.02 | **0.69 (−)** | 0.15 | 0.29 |

The relationship between the monthly relative abundance of the landscape units was also investigated, and the results are summarized in Table 7. It is remarkable that the relative abundance of landscape unit 0599 exhibits a very large negative correlation with the relative abundances of 0041, 0166, and 0313. This is expected from the results shown in Figure 7, since the vanishing of 0599 gives way to the increasing abundances of 0166 and 0313.

**Table 7.** Coefficient of determination between monthly relative abundance ($a_{LMS}$) of landscape units. Data in red are statistically significant at a 95% level (*p*-value < 0.05). Values with a very high coefficient of determination are marked in bold. The correlation is linear, being positive (+) or negative (-) as indicated. The symbol (exp) means that the correlation is exponential. Landscape units 0830 and 0041 coincide in only one month over the whole time period, and the correlation cannot be calculated.

| L | 0166 | 0313 | 0457 | 0599 | 0736 | 0830 |
|---|---|---|---|---|---|---|
| 0041 | 0.32 (+) | 0.02 (−) | 0.26 (−) | 0.42 (exp) (−) | 0.38 (−) | X |
| 0166 | | 0.15 (+) | 0.33 (−) | **0.81 (exp) (−)** | 0.09 (−) | 0.06 (−) |
| 0313 | | | 0.01 (−) | **0.73 (−)** | 0.31 (−) | 0.22 (−) |
| 0457 | | | | 0.02 (+) | 0.12 (−) | 0.13 (−) |
| 0599 | | | | | <0.01 | 0.10 (−) |
| 0736 | | | | | | 0.03 (+) |
| 0830 | | | | | | |

In order to address the issue of validity of the results because of the lack of data for some months, we then evaluated the results as shown above only for seasons 2012–2013 and 2013–2014, since for these seasons, all months are representative. These results are shown in Tables 8 and 9.

**Table 8.** Coefficient of determination between the monthly relative abundance ($a_{LMS}$) and the monthly mean albedo over Deception Island (<$\alpha$>) and the monthly mean air temperature from the GdC AWS ($T_{air}$) for seasons 2012–2013 and 2013–2014. Data in red are statistically significant at a 95% level (*p*-value < 0.05). Values with a very high coefficient of determination are marked in bold. The correlation is linear in all the cases, being positive (+) or negative (−) as indicated.

| L | 0041 | 0166 | 0313 | 0457 | 0599 | 0736 | 0830 |
|---|---|---|---|---|---|---|---|
| <$\alpha$> | 0.53 (−) | **0.61 (−)** | **0.89 (−)** | 0.02 | **0.82 (+)** | **0.66 (+)** | Only 2 data |
| $T_{air}$ | 0.25 | 0.40 | **0.75 (+)** | 0.04 | **0.86 (−)** | 0.49 | Only 2 data |

**Table 9.** Coefficient of determination between monthly relative abundance ($a_{LMS}$) of landscape units for seasons 2012–2013 and 2013–2014. Data in red are statistically significant at a 95% level (*p*-value < 0.05). Values with a very high coefficient of determination are marked in bold. The correlation is linear, being positive (+) or negative (−) as indicated. The symbol (exp) means that the correlation is exponential. Correlations with 0830 were not calculated due to the few data available.

| L | 0166 | 0313 | 0457 | 0599 | 0736 | 0830 |
|---|---|---|---|---|---|---|
| 0041 | **0.78 (+)** | <0.01 | 0.39 (−) | **0.93 (exp) (−)** | 0.05 | X |
| 0166 | | 0.04 | 0.18 | **0.86 (exp) (−)** | 0.15 | X |
| 0313 | | | 0.18 | **0.63 (−)** | 0.35 (−) | X |
| 0457 | | | | <0.01 | <0.01 | X |
| 0599 | | | | | <0.01 | X |
| 0736 | | | | | | X |
| 0830 | | | | | | |

Comparing Tables 6 and 8, we see that the correlations of monthly <$\alpha$> and $T_{air}$ with the monthly relative abundance obtained in the overall analysis follow the same behavior as those obtained for the 2012–2013 and 2013–2014 seasons. The correlations found have

the same sign in both cases. In the case of Table 6, landscape units 0599, 0166, and 0313 are the ones that seem to determine the albedo; and in the case of Table 8, it is units 0599, 0166, 0313, and 0736. However, 0736 is present in few months in the overall study period (Table 2), so it does not affect the general discussion. Regarding the correlation with $T_{air}$, landscape units 0599 and 0313 are the ones with the highest correlations in both cases. Discrepancies are observed in some cases, but the general picture is not affected. The same can be said when comparing the results of Tables 7 and 9. It is clear that the correlation of 0599 with the other landscape units is what determines the evolution of the landscape.

A more comprehensive view of the landscape evolution can be acquired by looking at the monthly evolution of the relative abundance in seasons 2012–2013 and 2013–2014 (Figures 8 and 9):

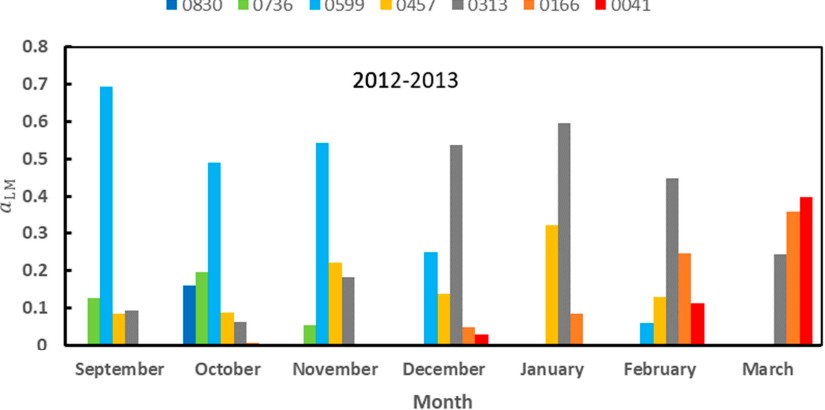

**Figure 8.** Relative abundance of each landscape unit for each month for the season 2012–2013 $\left(a_{LM,\,2012-2013}\right)$.

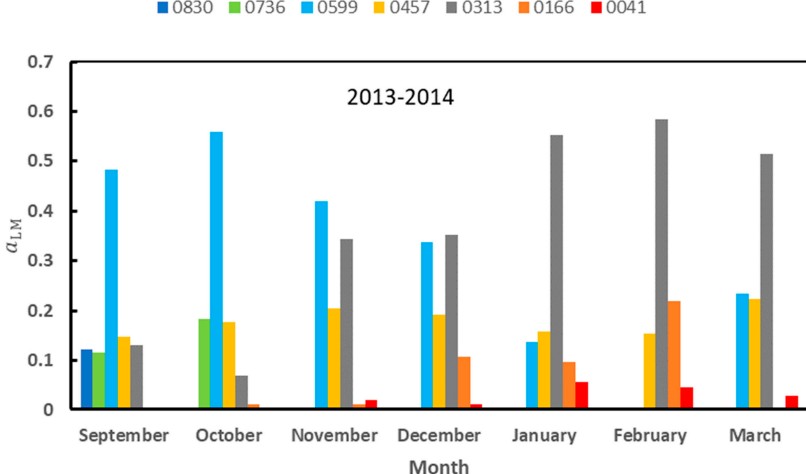

**Figure 9.** Relative abundance of each landscape unit for each month for the season 2013–2014 $\left(a_{LM,\,2013-2014}\right)$.

When we compare the trend of the mean monthly relative abundances in Figure 7 with those in Figures 8 and 9, we see that they follow the same general pattern: 0830 and 0736 vanish very soon in the season; 0599 follows a descending trend over the season; 0457 exhibits a nearly constant relative abundance over the season; and 0313, 0166, and 0041 take the space left by 0599, their relative abundance increasing in the second half of the season. From these comparisons, we conclude that the results obtained using the whole time span are not affected by the missing months.

### 3.4.3. Seasonal Relative Abundance

In Figure 10, we present the seasonal relative abundance $a_{LS}$ over the periods 2000–2001 to 2020–2021. There does not seem to be a clear trend in the seasonal evolution of the relative abundances. Nevertheless, we will show in the next section that there is a relationship between the relative abundances and meteorological variables.

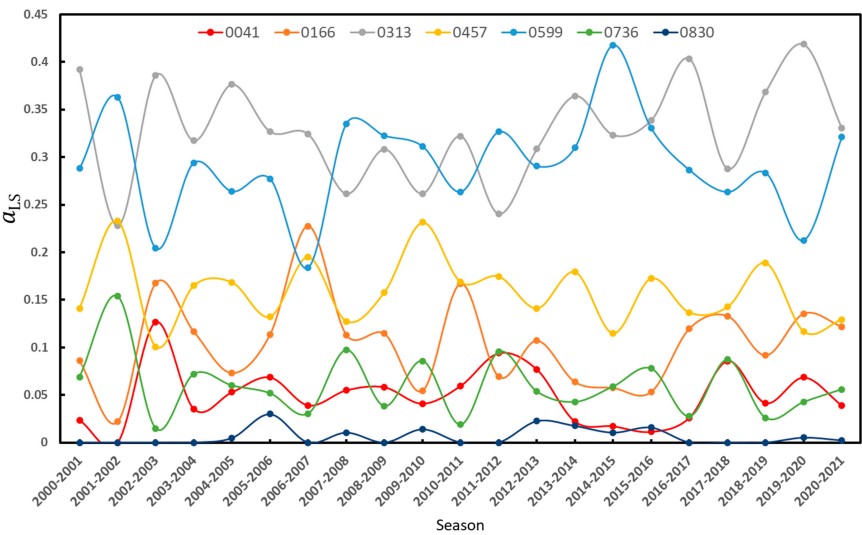

**Figure 10.** Seasonal relative abundance of each landscape unit in the period 2000–2021.

### 4. Discussion

The mean monthly albedo over Deception Island ($<\alpha>$) decreases steadily with increasing temperature, and the decreasing rate increases dramatically for temperatures above 272–273 K (Figure 5). A similar behavior has been found for snow albedo on other sites in Antarctica [24]. All the landscape units exhibit a significant correlation with $<\alpha>$, except 0457 and 0830 (Table 6). In the case of 0830, this is because the clean fresh snow disappears very soon in the season, probably due to the quick metamorphization and/or melting of the snow and because clean snow is very soon covered by ash carried by the wind. In the case of 0457, dirty snow, the cause is different. Dirty snow is present during the whole season with very small variation in the relative abundance. Landscape unit 0457 is the only landscape unit with this behavior (see Figure 7). This has an important impact on the description of albedo over Deception Island: it appears that ash covering snow, resulting in dirty snow, does not promote the melting of snow, contrary to what is expected based on physical arguments, since ash diminishes the albedo of snow. This could also be due to the fact that the layer of dirty snow is too thick to disappear completely. This point needs further investigation. Of all the meteorological variables measured at the GdC AWS, the mean monthly air temperature was the one exhibiting the highest correlation with the monthly abundance of a certain unit (Table 6), especially in the case of 0599, of which the relative abundance diminishes with increasing air temperature. On the other hand, the relative abundance of 0599 exhibits a high negative correlation with the monthly relative abundance of 0313 and 0166 (Table 7). We conclude that 0599 evolves into 0313, the evolution being driven by air temperature. Furthermore, solar radiation plays a fundamental role at the beginning of the season in the evolution of the relative abundance of 0599. A previous study of snow albedo decay over Livingston Island showed that albedo decay starts very soon in the season (in September), being driven by the solar energy flux density [25]. In Figure 11, we show the evolution of the mean daily solar energy flux density from the GdC AWS and the monthly relative abundance of 0599 from September to March between seasons 2005–2006 and 2020–2021. We can see that until December, increasing radiation coincides with a decreasing relative abundance of 0599.

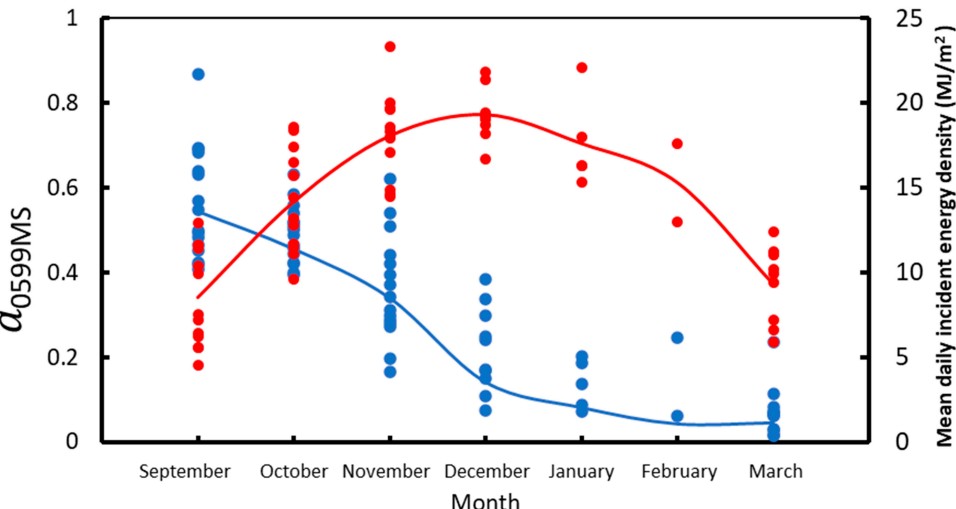

**Figure 11.** Monthly relative abundance of landscape unit 0599 (blue dots) and mean daily incident energy density (red dots) for each month between seasons 2005–2006 and 2020–2021. Each point represents a month of a season. The solid lines follow the mean value for each month and are given as a guide for the eye.

Wind is known to have an impact on snow albedo in two ways [26]. On the one hand, old snow can be exposed because of erosion, lowering the albedo [27,28]. In the case of Deception Island, this effect can be enhanced because of the exposition of dark bare soil. On the other hand, drifting snow grains reduce their size because of fragmentation and sublimation, inducing an increase of albedo [29]. Nevertheless, the correlation between snow cover and wind speed on Deception Island has been studied, and no correlation was found [14]. This is why wind was not taken into account in this work.

The results provide an accurate description of the land cover evolution during the season. Clean snow is only present at the beginning of the season, and it disappears very soon, melted or being covered by ash. As temperature and solar energy flux rise, snow melts. Pyroclasts beneath the snow surface, with high porosity, high air content, and large intergranular pore space, induce the fast percolation of melt water [5]. This results in the exposure of coarse pyroclasts and small outcrops around the exposed pyroclasts so that the rugged landscape of snow and pyroclasts (0599) becomes the dominant landscape unit. Solar radiation and the air temperature heat the exposed pyroclasts, which act as melting centers, promoting the transition from the rugged landscape (0599) to stripes of bare soil and snow (0313). Landscape units 0313 and 0041 coexist and share the space occupied earlier in the season by 0599. The transition from 0599 to 0313 and/or to 0041 is driven by the microtopography pattern, where some areas, with a rough soil surface, are prone to the transition from 0599 to 0313, while others that are flatter are prone to the transition from 0599 to 0041. A detailed study considering the topography is mandatory in order to fully understand in what cases 0599 evolves into 0313 or 0041. Landscape unit 0166 is actually a variant of 0041. Landscape unit 0166 is a very ephemeral cover that forms when a light snowfall falls on bare ground. An illustration of this mechanism is shown in Figure 12.

The distribution of landscape units may also explain the evolution of the permafrost active layer at the CALM site on Deception Island. The duration of the snow cover at the CALM-S site showed an increase from 2006 to 2014, especially with longer lasting snow cover in the spring and early summer [6]. In Figure 13, we show the seasonal relative abundance of 0599 and the combination of 0041 and 0166 from 2006–2007 to 2014–2015. The increase of 0599 and the decrease of 0041 and 0166 is consistent with these results, as presented in a previous study [6]. It is thus likely that this change in snow cover has occurred over the whole island. This trend in snow cover is also consistent with the deceleration in the glacial retreat observed on Livingston Island [30].

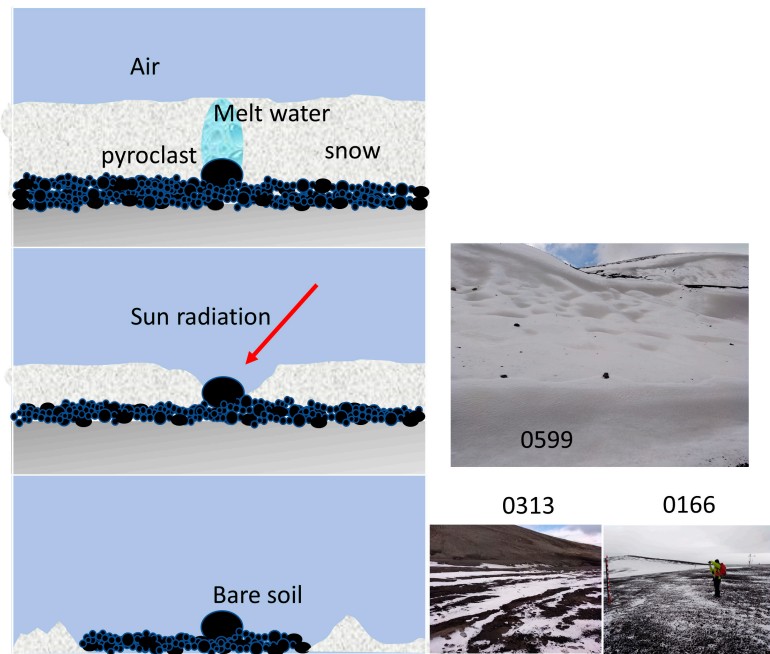

**Figure 12.** Mechanism proposed for the transition from landscape unit 0599 to either 0313 or 0166. Left panels describe the physical processes. Right panels show the corresponding landscape units. The arrow represents the solar radiation.

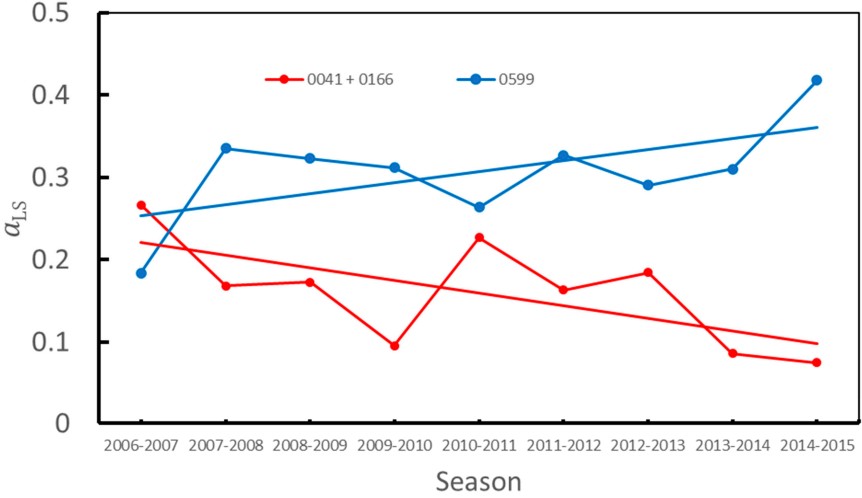

**Figure 13.** Seasonal relative abundance of landscape units 0041 and 0166 combined and 0599 in the period 2006–2015. Linear fits (solid straight lines) are shown as a visual guide of the trend. Blue dots represents the seasonal relative abundance of landscape unit 0599; red dots represent sum of the seasonal relative abundance of landscape units 0041 and 0166. Straight lines repereent linear fits of the relative abundance versus season, and they are given as a view guide.

Field albedo measurements over the landscape units combined with MODIS data with a spatial resolution of 500 m have been applied to different locations, and with different landscape units, in other studies. The selection of landscape units depends on the area under study, but in all the cases, they are chosen such that they have an area of the size of a MODIS pixel or larger. A study to validate MODIS data and study the variability of the albedo over a MODIS pixel was carried out in Canada [31]. In that study, the authors distinguished six landscape units and used them to investigate the influence of albedo variability inside each landscape unit on the observed MODIS albedo. The authors concluded that the accuracy of MODIS albedo depends on the albedo spatial variability

of the land cover, suggesting a fundamental limit to the root-mean-square error between MODIS albedo and in situ albedo close to 0.05 for snowfields and the tundra.

A study on the southwestern Greenland Ice sheet [4] defined four different landscape units, each one described by a normal distribution of shortwave albedo: clean ice, dirty ice, shallow streams, and cryoconite holes. The normal distributions were computed from field measurements along transects over the landscape units. The authors used these normal distributions along with information of their relative abundances to compute the albedo distribution over the melt season and compare it with the albedo distribution obtained from MODIS data. They found an increase of 51% in the surface melt in the transition from an albedo distribution dominated by dark pixels to an albedo distribution dominated by brighter pixels, with each distribution having a different relative abundance of the landscape units. The authors obtained the relative abundances of each landscape unit at different times during the melting season from a previous field survey [32]. The novelty of our approach lies in the fact that we have estimated the relative abundances of each landscape unit from the fit to satellite data. This opens the possibility of carrying out this kind of study in areas with the landscape units used in this study.

The seasonal evolution of albedo from the point of view of landscape units has also been studied at Taylor Valley (Mc Murdo Dry Valleys, Antarctica) [33]. In this work, the authors classify the landscape in three landscape units (glacier, lake, and soil) from true color images collected using a camera carried by a helicopter. They obtained that glacier and lake landscape units have a greater albedo variability than soil. This is due to sediment cover in ice-covered lakes and to roughness and debris over glaciers. They conclude that wind-driven snow and sediment redistribution are the driving factors of the spatiotemporal evolution of albedo. Compared to our research, we have used more landscape unit types, and we have been able to establish how and why some landscape units evolve into others, which is the critical point to understand the albedo evolution.

The role of surface cover in the evolution of glaciers has been noted in the case of cryoconite holes, which play an important role in the decrease of glacier albedo and in the promotion of melting [34].

The long-term evolution of albedo over Antarctica has also been investigated [35]. According to these authors, Deception Island should be included in a region of Antarctica where a slow decline of albedo was observed in the period 1983–2009. We have not observed a clear trend in the relative abundances of landscape units, except in the case of 0599 and 0041 + 0166 in the period 2006–2015 (Figure 13), which would cause not a decrease but an increase in albedo. In this respect, we must emphasize the importance of regional conditions that some studies at a global scale tend to miss.

**5. Conclusions**

We have devised a procedure to describe the evolution of the landscape over large areas and we have tested and applied it to an Antarctic site. The site chosen was Deception Island, which exhibits a seasonal changing landscape consisting of snow and bare soil. The procedure is based on combining broadband albedo data from a spaceborne sensor (MODIS) and from field surveys to obtain the monthly abundance of previously chosen landscape units. The landscape units were chosen following expert criteria and having a typical size of the order of the spatial resolution of the spaceborne sensor (MODIS). Each landscape unit was described by a normal distribution of broadband albedo, which was obtained from field measurements over selected sampling sites. The data were conveniently filtered to assure that the albedo variations over a sampling site were due to surface changes. We also used MODIS MCD43A3 (C6) Black Sky Albedo (BSA) to obtain the monthly mean albedo. The histograms of the MCD43A3 BSA are fitted to a linear combination of the normal distributions of the landscape units, and the coefficients of the linear combination are interpreted as the abundance of the landscape units. The results show that the evolution of the landscape is driven by the solar irradiance at the beginning of the season (September and October) and by the mean air temperature. The correlation between the abundances

provides information on how some landscape units evolve into others. In the case under study, a landscape unit consisting of a rugged landscape composed of dirty snow, with dispersed pyroclasts and small rocky outcrops, evolves into bare soil and a mixture of snow and bare soil patches. Pyroclasts beneath the surface, with high porosity, high air content, and large intergranular pore space, induce the fast percolation of snowmelt; this provokes the exposition of coarse pyroclasts and small outcrops. Solar radiation and air temperature heat the exposed pyroclasts which act as melting centers, promoting the transition. The transition is driven by the microtopography patterns. Ash on the snow surface does not seem to promote snowmelt.

**Author Contributions:** Conceptualization, J.F.C. and S.F.; methodology, J.F.C. and J.O.; software, R.M. and J.O.; formal analysis, J.F.C., J.O. and R.M.; investigation, J.F.C. and S.F.; resources, J.F.C., J.O., R.M. and S.F.; data curation, J.F.C., R.M. and S.F.; writing—original draft preparation, J.F.C., J.O. and C.R.; writing—review and editing, all authors.; visualization, J.F.C., J.O. and A.C.-P.; supervision, J.F.C. and S.F.; project administration, J.F.C. and S.F.; funding acquisition, J.F.C., J.O., F.N., M.Á.d.P. and S.F. All authors have read and agreed to the published version of the manuscript.

**Funding:** This research was funded by This work was supported by the Spanish Ministry of Science and Innovation under grants PID2021-127060OB-I00, PID2020-113051RB-C31, CTM2017-84441-R, and CTM2014-52021-R. The work of Alejandro Corbea-Pérez was supported by the Ph.D. Grant: "Severo Ochoa" from the Government of the Principality of Asturias [BP17-151].

**Data Availability Statement:** Data is unavailable due to privacy.

**Acknowledgments:** The authors thank the assistance of the technical staff of Spanish Antarctic Stations Juan Carlos I and Gabriel de Castilla during the 2017–2018 and 2018–2019 campaigns.

**Conflicts of Interest:** The authors declare no conflicts of interest.

## Appendix A

The MODIS mean monthly albedo ($<\alpha>$) was only calculated for months which met the criteria described in Section 2.4.2 to be classified as representative of Deception Island. The full list of months meeting these criteria comprises the ones marked with an X in Table A1.

**Table A1.** Months labeled as representative for the calculation of the monthly mean albedo $<\alpha>$. X = representative; blank = not representative.

| Season | S | O | N | D | J | F | M |
|---|---|---|---|---|---|---|---|
| 2000–2001 | X | | X | | | | |
| 2001–2002 | X | | | | | | |
| 2002–2003 | X | | | X | X | | X |
| 2003–2004 | X | X | X | | X | | X |
| 2004–2005 | X | X | X | X | | | X |
| 2005–2006 | X | X | X | X | | | X |
| 2006–2007 | X | X | X | X | | X | X |
| 2007–2008 | X | X | X | X | X | | X |
| 2008–2009 | X | X | X | | | | X |
| 2009–2010 | X | X | X | | | | X |
| 2010–2011 | X | X | X | X | X | | X |
| 2011–2012 | X | X | X | X | | | X |
| 2012–2013 | X | X | X | X | X | X | X |
| 2013–2014 | X | X | X | X | X | X | X |
| 2014–2015 | X | X | | | X | | X |

**Table A1.** *Cont.*

| Season | S | O | N | D | J | F | M |
|---|---|---|---|---|---|---|---|
| 2015–2016 | X | X | X | X | X | | X |
| 2016–2017 | X | X | X | X | X | | X |
| 2017–2018 | X | X | X | X | | X | X |
| 2018–2019 | X | X | X | X | | X | X |
| 2019–2020 | X | X | X | X | | X | X |
| 2020–2021 | X | X | X | X | | | X |

**Appendix B**

Sensitivity of the portable albedometer. Effect of clouds on field albedo measurements.

For the purpose of studying the sensitivity of the portable albedometer, we carried out five experiments:

Experiment A.—Standing still on bare soil. Pyranometers carried on the shoulders. 31 January 2019. Close to the AWS at JCI station. Overcast sky, 327 samples (27 min).

Experiment B.—Standing still on bare soil. Pyranometers carried on the shoulders. 31 January 2019. Close to the AWS at JCI station. Overcast sky. Very stable incident irradiance, 73 samples (6 min).

Experiment C.—Standing still on bare soil. Pyranometers carried on the shoulders. 1 February 2019. Close to the AWS at JCI station. Clear sky with clouds, large fluctuations in incident irradiance, 367 samples (31 min).

Experiment D.—Standing still on clean snow. Pyranometers carried on the shoulders. 25 January 2019. Close to the AWS at Hurd Glacier, Livingston Island. Clear sky. Very stable incident irradiance, 100 samples (8 min).

Experiment E.—Standing still on clean snow. Pyranometers carried on the shoulders. 1 February 2019. Close to the AWS at Hurd Glacier, Livingston Island. Overcast sky, 120 samples (10 min).

The surface cover at the AWS close to JCI station is bare soil between January and March; the surface cover at the AWS on Hurd Glacier is snow or ice all year round. Both AWSs provide the incident irradiance, reflected irradiance, and albedo every 10 min.

(1)    Estimation of fluctuations due to incident irradiance fluctuations

We analyze the incident irradiance, the reflected irradiance, and the albedo measured with the portable albedometer from Experiments A, B, C, D, and E. The data from the AWSs are not used at this point. The results are shown in Table A2.

**Table A2.** Results of the sensitivity experiments A, B, C, D, and E run in the proximity of the AWSs. Exp means experiment. CV = coefficient of variance calculated as the ratio of the standard deviation to the mean.

| Exp | | Incident (W/m$^2$) | Reflected (W/m$^2$) | Albedo |
|---|---|---|---|---|
| | Mean | 566 | 59 | 0.104 |
| A | Standard deviation | 34 | 3 | 0.003 |
| | CV (%) | 6 | 5 | 3 |
| | Mean | 556 | 59 | 0.106 |
| B | Standard deviation | 5 | 2 | 0.002 |
| | CV (%) | 1 | 3 | 2 |
| | Mean | 568 | 62 | 0.108 |
| C | Standard deviation | 160 | 20 | 0.007 |
| | CV (%) | 28 | 3 | 7 |

**Table A2.** *Cont.*

| Exp | | Incident (W/m²) | Reflected (W/m²) | Albedo |
|---|---|---|---|---|
| | Mean | 821 | 440 | 0.540 |
| D | Standard deviation | 5 | 3 | 0.004 |
| | CV (%) | <1 | <1 | <1 |
| | Mean | 466 | 266 | 0.571 |
| E | Standard deviation | 31 | 17 | 0.006 |
| | CV (%) | 7 | 6 | 1 |

The results show that:

(a)  As long as the coefficient of variance (CV) of the incident irradiance is below 7%, the standard deviation of albedo is below 0.003 for bare soil (Experiments A and B) and below 0.004 for snow (Experiment D). For other cover types consisting of a mixture of bare soil and snow, we assume values in between these two values. This means that when working with a dataset with a CV of the incident irradiance of 6% or less, variations of albedo above 0.003 for bare soil and above 0.004 for snow can be attributed to changes in the surface.

(b)  According to Experiments B and D (both correspond to very stable incident irradiance), we see that the noise of the portable albedometer (internal noise of the pyranometers plus noise caused by uncontrolled tilt of the pyranometers due to tiredness of the researcher or tilting of the body of the researcher over time, like, for example, when standing on snow) causes variations in albedo below 0.004. Variations above 0.004 under stable illumination must be caused by changes in the surface.

(2)  Estimation of biased uncertainties caused by the experimental set-up (pyranometers model, influence of the body of the researcher carrying the pyranometers).

In this case, the results obtained with the portable albedometer in Experiments D and E were averaged and compared to those acquired by the AWSs. We take the data from the AWS as truth. Unfortunately, the AWS at JCI does not provide any data for the dates of Experiments A, B, and C, so the biased uncertainties could only be tested on snow. The results are shown in Table A3 below.

**Table A3.** Results of Experiments D and E. Ratio is the ratio of the data from the portable albedometer to the data from the AWS. Exp means experiment. The relative difference is calculated as (Portable albedometer—AWS)/AWS.

| Exp | | Incident (W/m²) | Reflected (W/m²) | Albedo |
|---|---|---|---|---|
| | AWS | 847 | 513 | 0.606 |
| D | Portable albedometer | 820 | 441 | 0.538 |
| | Ratio | 0.968 | 0.860 | 0.888 |
| | Relative Difference | −0.032 | −0.140 | −0.112 |
| | AWS | 554 | 354 | 0.639 |
| E | Portable albedometer | 488 | 278 | 0.570 |
| | Ratio | 0.881 | 0.785 | 0.892 |
| | Relative Difference | −0.119 | −0.215 | −0.108 |

The maximum difference in albedo is −11.2% (Experiment D). The relative difference is larger for the reflected irradiance than for the incident irradiance due to the blocking of light by the body of the researcher. On the other hand, although the difference in the reflected irradiance can be as high as 21.5% (Experiment E), this is compensated by the difference in the incident irradiance.

(3)  Effect of clouds

Because of time constraints and logistics during the Antarctic campaign, some of the field albedo measurements had to be performed under cloudy conditions. While snow albedo is very sensitive to clouds, that of bare soil is not. Clouds change the spectral composition of the incident radiation due to the strong absorption in the infrared part of the spectrum. Snow spectral albedo exhibits a great dependence on wavelengths, while soil spectral albedo depends very smoothly on wavelengths. Snow-covered landscape units (0457, 0599, 0736, and 0830) were corrected for cloudiness following the method proposed in a previous work [36]. The method consists of calculating the albedo under a clear sky from the actually measured albedo as:

$$\alpha_{clear} = \alpha_{cloud} + 0.05(n - 0.5) \tag{A1}$$

where $\alpha_{clear}$ is the albedo that would have been measured under a clear sky, $\alpha_{cloud}$ is the actually measured albedo, and n is the cloud index ($n = 1$ means a completely overcast sky, $n = 0$ means a completely clear sky). Equation (A1) can only be applied on snow-covered surfaces. The cloud index can be calculated from the cloud transmittance ($T$) and the altitude above sea level of the observation site (h in m) using the relation:

$$T = 1 - An^2 e^{-Bh} \tag{A2}$$

with $A = 0.78$ and $B = 0.00085$ m$^{-1}$, empirically derived constants.

The value of T is calculated assuming that

$$T = \frac{E(cloud)}{E(clear)} \tag{A3}$$

where *E(cloud)* is the actually measured irradiance and *E(clear)* is the irradiance that would have been measured under clear-sky conditions. E(clear) depends on the Sun Zenith Angle *(SZA)*. To calculate *E(clear)* at the time of the acquired *E(cloud)*, we used the measured irradiance on the closest date with clear-sky conditions. In our case, this happened on February 16, 2019. *E(clear)* at any time of the day was obtained by fitting hourly irradiance to the SZA:

$$E(clear) = a(cos(SZA))^b \tag{A4}$$

From the fit, we obtained a = 1088 W/m$^2$ and b = 1.7, with a coefficient of determination R$^2$ = 0.97.

The results of the correction are shown in Table A4.

**Table A4.** Mean albedo and standard deviation of landscape units L = 0457, 0599, 0736, and 0830 before (measured) and after (corrected) the cloudiness correction. R. D. = relative difference measured as (Measured—Corrected)/Corrected.

| | **L** | | | | | | | |
| | **0457** | | **0599** | | **0736** | | **0830** | |
| | $<\alpha>$ | $\Sigma$ | $<\alpha>$ | $\sigma$ | $<\alpha>$ | $\sigma$ | $<\alpha>$ | $\sigma$ |
|---|---|---|---|---|---|---|---|---|
| Corrected | 0.437 | 0.016 | 0.578 | 0.041 | 0.722 | 0.014 | 0.805 | 0.016 |
| Measured | 0.457 | 0.018 | 0.599 | 0.040 | 0.736 | 0.013 | 0.830 | 0.016 |
| R. D. | 0.05 | 0.10 | 0.04 | −0.02 | 0.02 | −0.07 | 0.03 | 0.00 |

The effect of clouds is an increase in the albedo mean between 2% and 5% and a variation in the standard deviation between −7% and 10%. We assume that the correction for bare soil due to clouds is negligible. The correction for mixtures of bare soil and snow cannot be quantified, but we assume that it is below that obtained for surfaces completely covered by snow.

In summary, we have calculated the effect of the experimental set-up and clouds on snow-covered surfaces. The effect of the experimental set-up is a decrease in the measured albedo with respect to that from the AWSs, with a maximum decrease of −11.2%. On the other hand, clouds produce an increase in the measured albedo with respect to the cloud-free one in the range from 2% to 5% and an uncertainty in the standard deviation in the range from −7% to 10%. It has been impossible to calculate the correction due to these factors on the rest of the landscape units: on the one hand, the AWS at JCI (bare soil) does not provide data for the dates of the experiments; on the other hand, a procedure to correct albedo measurements for the effect of clouds is only known for a snow-covered surface. We think it would not be a good procedure to correct some surfaces and not others.

(4)    Impact of corrections

To estimate the impact of bias and clouds on the results, we assume that landscape units 0041, 0166, and 0313 are only affected by the bias due the experimental set-up, and landscape units 0457, 0599, 0736, and 0830 are affected by bias and by cloudiness. We assume that bias provokes a decrease in albedo of 11.2% (the maximum observed) for all the landscape units, while clouds provoke the increase in albedo and the variation in the standard deviation shown in Table A4. Taking account all of these corrections, we assume that the corrected landscape units are the ones shown in Table A5 (the identification codes are the same as in the main body of the manuscript).

**Table A5.** Landscape unit code (*L*), mean albedo ($\mu_L$), and standard deviation ($\sigma_L$) of the normal distribution of each landscape unit after correction for cloudiness and bias due to the experimental set-up.

| *L* | Description | $\mu_L$ | $\sigma_L$ |
|---|---|---|---|
| 0830 | Clean fresh snow | 0.898 | 0.016 |
| 0736 | Clean old snow | 0.804 | 0.014 |
| 0599 | Rugged landscape of snow and pyroclasts | 0.647 | 0.041 |
| 0457 | Dirty snow | 0.488 | 0.016 |
| 0313 | Stripes of bare soil and snow | 0.348 | 0.080 |
| 0166 | Shallow snow and bare soil holes | 0.185 | 0.053 |
| 0041 | Bare soil | 0.046 | 0.009 |

We repeated the calculations of Section 2.4.3 with the new normal distributions. Let us denote by $a_{LMS}$ the monthly relative abundance obtained. We calculated the correlation of <$\alpha$> and $T_{air}$ with $a_{LMS}$ and the correlation between the $a_{LMS}$ for the 2012–2013 and 2013–2014 seasons together. These two seasons were chosen because all the months attained the label of representative. The results are shown in Tables A6 and A7 below, and they have to be compared with those of Tables 8 and 9 in the main body of the article. Although the values of the coefficients of correlation change, the main results remain unchanged. The signs of the correlations are the same with and without correction. The landscape units exhibiting a higher correlation with <$\alpha$> and $T_{air}$ are the same with and without correction.

**Table A6.** Coefficient of determination between the monthly relative abundance ($a_{LMS}$) and the monthly mean albedo over Deception Island (<$\alpha$>) and the monthly mean air temperature from the GdC AWS ($T_{air}$) for seasons 2012–2013 and 2013–2014. Data in red are statistically significant at a 95% level (*p*-value < 0.05). Values with very high coefficient of determination are marked in bold. The correlation is linear in all the cases, being positive (+) or negative (−) as indicated.

| *L* | 0041 | 0166 | 0313 | 0457 | 0599 | 0736 | 0830 |
|---|---|---|---|---|---|---|---|
| <$\alpha$> | 0.44 (−) | 0.75 (−) | 0.63 (−) | 0.03 | 0.74 (+) | 0.67 (+) | Only 2 data |
| $T_{air}$ | <0.01 | 0.18 | 0.81 (+) | 0.20 | 0.55 (−) | 0.49 | Only 2 data |

**Table A7.** Coefficients of determination between monthly relative abundances ($a_{LMS}$) of landscape units for seasons 2012–2013 and 2013–2014. Data in red are statistically significant at a 95% level ($p$-value < 0.05). Values with very high coefficient of determination are marked in bold. The correlation is linear, being positive (+) or negative (−) as indicated. The symbol (exp) means that the correlation is exponential. Landscape units 0830 and 0041 coincide in only one month over the whole time period, and the correlation cannot be calculated.

| $L$ | 0166 | 0313 | 0457 | 0599 |
|---|---|---|---|---|
| 0041 | **0.64 (+)** | <0.01 | 0.13 | **0.64 (exp) (−)** |
| 0166 | | 0.12 | 0.22 | **0.71 (exp) (−)** |
| 0313 | | | 0.18 | **0.67 (−)** |
| 0457 | | | | 0.10 |

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
