# Peer review of "Spatiotemporal Evolution of the Land Cover over Deception Island, Antarctica, Its Driving Mechanisms, and Its Impact on the Shortwave Albedo"

_remotesensing, doi:10.3390/rs16050915_

Round 1

Reviewer 1 Report

Comments and Suggestions for Authors

Goal of the  authors  is to describe the albedo evolution of albedo in a islad othe the Antartica Peninsula, comparing AWS measurements,  man-hold measurements and satellite analysis. The final aim is to relate punctual albedo and spacial extension, and discover relation between solar fluxes, albedo and temperature. Authors claim only uncovered pyroclasts act as melting center, while ash seems not to have a strong influence on albedo.

general comments:

the paper seems to focus on the relation between land cover broadband refectace characteristics and their evolution, focusing on solar irradiance and temperature. Could the authors try to comment the effect of wind on snow cover and conseguently  the surface albedo?  
In addiction, could the authors comment about the footprint of the "field data?"  which is the height of the albedometer? 1.5 meter?   for example :. Levy, C.R.; Burakowski, E.; Richardson, A.D. Novel Measurements of Fine-Scale Albedo: Using a Commercial Quadcopter to Measure Radiation Fluxes. Remote Sens. 2018, 10, 1303. https://doi.org/10.3390/rs10081303

paragraph 3.4.2: can authors comment the significance of the GdC AWS and the whole island?

minor comments
the lines 82-85 in my opinion are not so relevant to the scope of the paper.
line 182: equration => equation
line 182: please describe the quantity "d"
line 191 : form => from

legend figure 4: please remark the scale difference in the frequency scale

figure 6: please add the y label in the figure ( and also remark the scale difference)

Author Response

We would like to thank the reviewer for his comments. They have been crucial to improve the quality of the manuscript.

We have answered all this questions.

Reviewer’s questions are underlined.

Our answers are in italics.

Corrections in the manuscript have been added in red.

Reviewer 1

Goal of the  authors  is to describe the albedo evolution of albedo in a islad othe the Antartica Peninsula, comparing AWS measurements,  man-hold measurements and satellite analysis. The final aim is to relate punctual albedo and spacial extension, and discover relation between solar fluxes, albedo and temperature. Authors claim only uncovered pyroclasts act as melting center, while ash seems not to have a strong influence on albedo.

general comments:

the paper seems to focus on the relation between land cover broadband refectace characteristics and their evolution, focusing on solar irradiance and temperature. Could the authors try to comment the effect of wind on snow cover and conseguently  the surface albedo?  

We have added a new paragraph to explain why we did not take into account the wind:

Wind is known to have an impact on snow albedo in two ways [26]. On the one hand, old snow can be exposed because of erosion, lowering the albedo [27,28]. In the case of Deception Island, this effect can be enhanced because of the exposition of dark bare soil. On the other hand, drifting snow grains reduce their size because of fragmentation and sublimation, inducing an increase of albedo [29]. Nevertheless the correlation between snow cover and wind speed on Deception Island has been studied and no correlation was found [14] This is why wind was not taken into account in this work.

In addiction, could the authors comment about the footprint of the "field data?"  which is the height of the albedometer? 1.5 meter?   for example :. Levy, C.R.; Burakowski, E.; Richardson, A.D. Novel Measurements of Fine-Scale Albedo: Using a Commercial Quadcopter to Measure Radiation Fluxes. Remote Sens. 2018, 10, 1303. https://doi.org/10.3390/rs10081303.

A new paragraph has been added to the manuscript:

The footprint of the field data is estimated to be a circle with a diameter ten times de height of the sensors [16]. In our case, the height of the sensors is 1,5 m, so the footprint of the field data is a circle of a diameter of 15 m. For each landscape unit, we checked that the samples were distributed according to a normal distribution. This is especially critical in the case of landscape unit 0313: if the sensors were placed too low, we would sample bare soil and snow separately, yielding a binomial distribution.   

paragraph 3.4.2: can authors comment the significance of the GdC AWS and the whole island?

We explain this in a new paragraph:

Some variables exhibit a great dependence on topography. Wind velocity, wind di-rection and air temperature have been measured at AWSs on Deception Island and on King George Island, located 120 km apart [14]. The results show that wind direction and velocity on Deception island follow a very different behaviour to that measured at King George Island, while daily air temperature at Deception Island follows the same temporal trend as daily air temperatures measured at King George Island. According to these re-sults, we think that we can take the air temperature measured at the GdC AWS as repre-sentative of the whole island.

minor comments
the lines 82-85 in my opinion are not so relevant to the scope of the paper.

DONE: they have been deleted

line 182: equration => equation

DONE

line 182: please describe the quantity "d"

A new paragraph explains this:

In the calculation of the albedo, the incident radiance can be divided into a direct compo-nent with angles θ,φ (Edir(θ,φ)) and a diffuse component Ldiff and assuming that Ldiff is iso-tropic, we define the fraction of diffuse radiation as d = (πLdiff/Edir) [19].

 line 191 : form => from

DONE

legend figure 4: please remark the scale difference in the frequency scale

DONE

figure 6: please add the y label in the figure ( and also remark the scale difference)

DONE

Reviewer 2 Report

Comments and Suggestions for Authors

The authors used ground-based observations of air temperature, incident solar radiation and surface albedo together with satellite-based measurements of surface albedo to describe the seasonal evolution of surface types over Deception Island in Antarctica. They applied a promising approach to estimate the contributions of the different landscapes within a MODIS pixel.

In general, this study is easy to read, but the quality of the illustrations should be improved. The labeling is quite small (e.g. Figure 6). I can recommend this manuscript for publication in Remote Sensing. However, some comments should be considered beforehand.

11. Can you validate your approach to estimate the contribution of the different landscape units within a MODIS pixel with high resolution land cover data? Another possibility would be to create artificial low resolution albedo data from mixtures of landscapes and try to reproduce the original mixture.

22. Why did the authors use the air temperature as a parameter affecting the albedo. The surface skin temperature (satellite data) might be a more suitable parameter. This is used in albedo modelling. Perhaps data of snow thickness is also available?

33. The authors should additionally discuss the effect of the solar zenith angle and topography / surface roughness on surface albedo. The study area is not flat and the solar zenith angle changes naturally.

44. The conclusion that air temperature and incident irradiance are responsible for the landscape changes is not surprising. In fact, they are linked.

55. In discussing the validity of the results the authors write: “... overall analysis the same behaviour of those obtained for 2012-2013 and 2012-2014 seasons” (p13l446), or “the main results remain unchanged” (p23l749). What is an acceptable range? What are the criteria? A kind of measure is missing.

66. The role of clouds on surface albedo is described as “great extent” (p6l198). It is claimed that the “field albedo data must be corrected for the effect of clouds” (p6l205).  Nevertheless, the authors do not apply a correction in the main body of the manuscript. They show some corrected results in the appendix and argue that “it would not be a good procedure to correct some surfaces and not others” (p22l729), even though it is mentioned in Section 2.4.1 that “soil albedo is known to be insensitive to changes in incident irradiance”. Please reconsider your strategy, as cloud correction is really important in my view.

Minor comments:

P2l61: “The reflectance, expressed in the albedo”. Better omit “reflectance” here and in line 63, as they are two different quantities.

P2l73: “...albedo can increase because of” – The sentence sounds odd, as the area covered by snow is mostly due to new fresh snow.

P2l79: “The advantage of the method ...” – I do not think that field measurements are easy to carry out in this harsh environment.

P2l82: I would not mention geoengineering here. It should not be a motivation of this study. Moreover, the applied method may help to analyse and understand small scale surface evolution in the polar regions. The authors should rather give an outline of their study at the end of the introduction.

P3l119: What is meant by calibration of “field albedo measurements”? Please elaborate. Some information should be given in the main body of the manuscript.

P3L120: Please clarify that no albedo data are needed from the GdC AWS. I was puzzled because in line 124 it reads: “data from GdC AWS are going to be used to study the evolution of albedo and landscape units...”

Table 1: Give measurement uncertainties.

P5l182: What is the reason for that small relative difference between WSA and BSA?

P6l208: “Fluctuations due to incident irradiance fluctuations ...” Change wording. Further, also cloudiness (point 3) contributes to changes of the irradiance.

Sec. 2.4.3: Think about to demonstrate the method using a graphical example.

Table 3: Could be described better. The number of months with presence of each landscape is not discussed.

P9l327: “Radiation data has been measured ...” The last section is not really needed here. Is this shown only to document the cloudiness for the timeframe of the used observations?

Sec. 3.3 is not well motivated. I would move it after 3.4.1.

Appendix: I would not call the sensitivity study a calibration. As I understand, the absolute numbers of the field albedo measurements are not adjusted. The absolute calibration was probably performed by the manufacturer of the pyranometers.

P21l676: “On the other hand, although ...” Why give  both pyranometers  lower values than the AWS data set?

Equ. A1: albedo_clear and albedo_cloud seem to be mixed up.

Equ. A2: Give the unit of the altitude.

Author Response

We would like to thank the reviewer for his comments. They have been crucial to improve the quality of the manuscript.

We have answered all this questions.

Reviewer’s questions are underlined.

Our answers are in italics.

Corrections in the manuscript have been added in blue.

The authors used ground-based observations of air temperature, incident solar radiation and surface albedo together with satellite-based measurements of surface albedo to describe the seasonal evolution of surface types over Deception Island in Antarctica. They applied a promising approach to estimate the contributions of the different landscapes within a MODIS pixel.

In general, this study is easy to read, but the quality of the illustrations should be improved. The labeling is quite small (e.g. Figure 6).

A new figure has been included in the manuscript.

 I can recommend this manuscript for publication in Remote Sensing. However, some comments should be considered beforehand.

  1. Can you validate your approach to estimate the contribution of the different landscape units within a MODIS pixel with high resolution land cover data? Another possibility would be to create artificial low resolution albedo data from mixtures of landscapes and try to reproduce the original mixture.

In this work we work with the concept of “landscape unit”, which should not be confused with surface cover. Some landscape units are pure, i.e., they consist of a single surface cover, like 0830, 0736, 0457 and 0041. Other are a mixture of surface covers, like 0599, 0313 and 0166, which consist of different kinds of snow and bare soil mixtures. The landscape units were chosen such that their typical size is that of a MODIS pixel. Had we used another sensor with a different spatial resolution, the landscape units would have been different. We could have added a high resolution image to show the spatial extent of each of the landscape units. Some landscape units can only be distinguished using ultra high resolution imagery with an UAV of in-situ photographs (in-situ photographs are shown in the manuscript, Figure3). For example, landscape units 0599 and 0457 can only be distinguished using a sensor capable of detecting small rocky outcrops. The main advantage of our procedure is that we can associate albedo variations at a 500-m pixel level to changes in the range of a few meters that induce albedo changes.

  1. Why did the authors use the air temperature as a parameter affecting the albedo. The surface skin temperature (satellite data) might be a more suitable parameter. This is used in albedo modelling. Perhaps data of snow thickness is also available?

We are interested in studying the influence of the atmosphere on the evolution of the landscape. Surface temperature could have been obtained from MODIS, for example. It is worth noting here that a lot of work is focused in obtaining air temperature from surface temperature (which is the temperature directly measured by spaceborne sensors). We think that air temperature is the key parameter here.

Furthermore, here again, we must take into account that in this work we adopt the point of view of landscape units. Some of the landscape units are a mixture of surface covers, each surface cover might have a different surface temperature. Think, for example, of landscape units 0599, 0313 and 0166. Just as the albedo from different surface covers can be aggregated as the weighted average (by the relative abundance) of each type of cover, this is not obvious in the case of temperature. So, it is not clear how the surface temperature is associated to the composition of a landscape unit. This is another point to explain why we decided to work with air temperature.

  1. The authors should additionally discuss the effect of the solar zenith angle and topography / surface roughness on surface albedo. The study area is not flat and the solar zenith angle changes naturally.

This is a very interesting point. In a previous paper (Snow Albedo Seasonal Decay and Its Relation With Shortwave Radiation, Surface Temperature and Topography Over an Antarctic ICE Cap January 2021, IEEE Journal of Selected Topics in Applied Earth Observations and Remote Sensing PP(99):1-1) we demonstrated that topography is only important when the snow cover is homogeneous, which is not the case for most of the season over Deception Island, because the snow cover becomes patchy very soon in the season. We think, as stated in the paper, that topography is essential to understand why some landscape units appear instead of others. But this is out of the scope of this paper, and needs further research, as pointed out in the manuscript.

  1. The conclusion that air temperature and incident irradiance are responsible for the landscape changes is not surprising. In fact, they are linked.

The point here is that bare soil consisting of dark material is ubiquitous. Dark soil absorbs a lot of radiation, the role of radiation being crucial at the beginning of the season (see Figure 11). For example, while the relative abundance of 0599 exhibits a high negative linear correlation with Tair (see Table 5, in which all the months are included), the relationship with solar radiation is not so straightforward. According to Table 5, increasing Tair induces a decrease of 0599. However, the relationship between the relative abundance of 0599 and the solar radiation is more complicated: while they seem to have a high negative correlation at the beginning of the season (from September to December) the relationship changes from December to March.

  1. 55. In discussing the validity of the results the authors write: “... overall analysis the same behaviour of those obtained for 2012-2013 and 2012-2014 seasons” (p13l446), or “the main results remain unchanged” (p23l749). What is an acceptable range? What are the criteria? A kind of measure is missing.

We have not been able to devise any kind of quantitative criteria. We have added a brief qualitative description.

Comparing Table 6 and Table 8 we see that the correlations of monthly <α> and Tair  with the monthly relative abundance obtained in the overall analysis follows the same behaviour as those obtained for 2012-2013 and 2013-2014 seasons. The correlations found have the same sign in both cases. In the case of Table 6, landscape units 0599, 0166 and 0313 are the ones that seem to determine the albedo, and in the case of Table 8 it is units 0599m 0166, 0313 and 0736. However, 0736 is present in few months in the overall study period (Table 2), so it does not affect the general discussion. Regarding the correlation with Tair, landscape units 0599 and 0313 are the ones with the highest correlation in both cases. Discrepancies are observed in some cases, but the general picture is not affected. The same can be said when comparing the results of Table 7 and 9. It is clear that is the correlation of 0599 with the other landscape units what determines the evolution of the landscape.

The signs of the correlations are the same with and without correction. The landscape units exhibiting a higher correlation with <α> and Tair is the same with and without correction.

  1. The role of clouds on surface albedo is described as “great extent” (p6l198). It is claimed that the “field albedo data must be corrected for the effect of clouds” (p6l205).  Nevertheless, the authors do not apply a correction in the main body of the manuscript. They show some corrected results in the appendix and argue that “it would not be a good procedure to correct some surfaces and not others” (p22l729), even though it is mentioned in Section 2.4.1 that “soil albedo is known to be insensitive to changes in incident irradiance”. Please reconsider your strategy, as cloud correction is really important in my view.

We think that cloud correction is very important if you want to predict the value of albedo at a given location on a given date with high accuracy. In this work, we demonstrate that the error in the albedo estimation does not affect the overall conclusions of the results. Moreover, the effect of clouds on snow albedo is well known and several algorithms have been proposed in the literature, like the one we use, but in the case of other types of surface covers the effect of clouds on albedo is not so well known. This is why, instead of correcting for the effect of clouds some landscape units (those made up of snow) and not others (soil and mixture of soil and clouds) we decided to work with the field data as-measured, and study the effect of cloud correction (in the surfaces where this correction can be done).

Minor comments:

P2l61: “The reflectance, expressed in the albedo”. Better omit “reflectance” here and in line 63, as they are two different quantities.

DONE

P2l73: “...albedo can increase because of” – The sentence sounds odd, as the area covered by snow is mostly due to new fresh snow.

We have rewritten the whole sentence. Now it reads:

On the other hand, albedo can increase if new fresh snow falls on old snow, like in a snowfall event, or if the area covered by snow increases.

P2l79: “The advantage of the method ...” – I do not think that field measurements are easy to carry out in this harsh environment.

We mean that compared with spectroscopic measurements using an ASD, for example, the measurements are easy to carry out because of the equipment.

The sentence has been rewritten as follows:

The advantage of the method presented lies in the fact that field measurements are easy to carry out, even in the harsh environment of Deception Island, since the equipment is rough, light and easy to transport over snowed and iced areas.

P2l82: I would not mention geoengineering here. It should not be a motivation of this study. Moreover, the applied method may help to analyse and understand small scale surface evolution in the polar regions. The authors should rather give an outline of their study at the end of the introduction.

DONE.

New text has been added at the end of the Introduction:

This work is organized as follows. In Section 2 we provide a description of the study area, the data used and the data processing. In Section 3 we present the results. In Section 4 we discuss the results obtained presenting the mechanism proposed to explain the evolution of the landscape over Deception Island and its driving mechanisms.

P3l119: What is meant by calibration of “field albedo measurements”? Please elaborate. Some information should be given in the main body of the manuscript.

A precise description of what the calibration is and why it is important is given in section 2.4.1 in the main body of the manuscript.

We have included the following words where the reviewer recommends:

The calibration consists in comparing the field data to data collected by a AWSs (taken as truth). This will allow us to evaluate the impact of several factors on the field data: the response of the sensors to changing incident irradiance, the body of the researcher carrying the sensors and clouds.

P3L120: Please clarify that no albedo data are needed from the GdC AWS. I was puzzled because in line 124 it reads: “data from GdC AWS are going to be used to study the evolution of albedo and landscape units...”

Following a comment below, we substituted the word “calibration” for “sensitivity” or “sensitivity analysis” throughout the manuscript.

We have added the following text:

Albedo data from AWSs is only needed for sensitivity analydis purposes. For the investigation of the driving mechanism of the evolution of the landscape units, we need incident radiation and air temperature as close as possible to the study area. These data are available at GdC AWS on Deception Island.

Table 1: Give measurement uncertainties.

DONE.

P5l182: What is the reason for that small relative difference between WSA and BSA?

Because the bidirectional reflectance distribution function depends very weakly on the direction of the incident radiation.

P6l208: “Fluctuations due to incident irradiance fluctuations ...” Change wording. Further, also cloudiness (point 3) contributes to changes of the irradiance.

We substituted the word “fluctuations” for “oscillations”.

Sec. 2.4.3: Think about to demonstrate the method using a graphical example.

Figure 6 is a graphical example.

Table 3: Could be described better. The number of months with presence of each landscape is not discussed.

A brief discussion of the result is now provided:

Only landscape units 0457 and 0313 are permanently present through the whole season. Clean fresh snow (0830) can only be observed in September and October, and even in those months its presence is rare. Old snow is ubiquitous is September and October, and disappears abruptly from November on. Landscape unit 0599 is permanently observed from September to November, its presence decaying softly the rest of the season. Bare soil (0041) can barely be observed at the beginning of the season, and it is covered by snowfall events (0166). As the season advances, bare soil becomes one of the dominant landscapes.

P9l327: “Radiation data has been measured ...” The last section is not really needed here. Is this shown only to document the cloudiness for the timeframe of the used observations?

This information is given to explain why there are fewer satellite data in January and February.

Sec. 3.3 is not well motivated. I would move it after 3.4.1.

We have motivated this section as follows:

In this work we intend to link albedo to landscape. We hypothesize that one the driving mechanisms of albedo evolution and landscape change is air temperature. It is worth gaining insight in the relationship between albedo and air temperature on Deception Island.

Appendix: I would not call the sensitivity study a calibration. As I understand, the absolute numbers of the field albedo measurements are not adjusted. The absolute calibration was probably performed by the manufacturer of the pyranometers.

We have substituted the word “calibration” for “sensitivity”. Now this paragraph reads as follows:

Sensitivity of the portable albedometer. Effect of clouds on field albedo measurements. 

For the purpose of studying the sensitivity of the portable albedometer, we carried out five experiments:

P21l676: “On the other hand, although ...” Why give  both pyranometers  lower values than the AWS data set?

This is because the body of the researcher covers part of the field of view of the sensors.

Equ. A1: albedo_clear and albedo_cloud seem to be mixed up.

Equ. A2: Give the unit of the altitude.

DONE

Reviewer 3 Report

Comments and Suggestions for Authors

This work studies how air temperature and solar radiation induce changes in the land cover over an Antarctic site. The authors use shortwave broadband albedo from a spaceborne sensor and field measurements to study the albedos and relative abundance of landscape units for seven landscape units, as well as a mechanism to describe the evolution of the landscape. My comments and suggestions are as follows.

In abstract, it is suggested to report the flied measurement period.

L299-300 and Table 2, The landscape units with the larger standard deviation are those that consist of a mixture of bare soil and snow (0599, 0313, and 0166). Pleas explain the reasons for the larger standard deviations, e.g., from the measurements?

L349-350, should be rewritten.

3. It is suggested to show the long-term variations of albedos for 7 landscapes, and discuss their variation reasons. There are some studies about albedos in the Antarctic, please make comparisons with them.

3. It is also suggested to make comparisons between the field measurement and satellite albedos.

Figure 10, please make changes and show all lines clearly.

Figure 11, should be rewritten, the meanings of blue dots and blue solid line are not clear.

It is suggested to report detailed information (e.g., location of sampling site, time periods, sky conditions, Cloudiness) about the field measurements and satellite data for the albedos in a table.

Author Response

We would like to thank the reviewer for his comments and ideas to improve the quality of the manuscript.

We have revised the paper and introduce some changes following the reviewer’s recommendations.

Reviewer’s question are underlined.

Our answeres are in italics.

New text added to the manuscript is in orange.

This work studies how air temperature and solar radiation induce changes in the land cover over an Antarctic site. The authors use shortwave broadband albedo from a spaceborne sensor and field measurements to study the albedos and relative abundance of landscape units for seven landscape units, as well as a mechanism to describe the evolution of the landscape. My comments and suggestions are as follows.

In abstract, it is suggested to report the flied measurement period.

DONE

L299-300 and Table 2, The landscape units with the larger standard deviation are those that consist of a mixture of bare soil and snow (0599, 0313, and 0166). Pleas explain the reasons for the larger standard deviations, e.g., from the measurements?

We have to distinguish between landscape unit and surface cover. Here we have, roughly speaking, two surface covers: bare soil and snow. The albedo of  “Pure” landscape units, consisting of a single surface cover, will only depend on the variations due to surface cover variations, that will be very small. On the other hand, the albedo of “mixed” landscape units, made up of different surface covers, will depend on the inherent variations of each surface cover plus on the amount of each surface cover sampled in a single sample.

We have added the pargraph:

This is the expected result. In this work we deal with the concept of landscape unit. Some landscape units are made of a single surface cover (0041, 0457, 0736, and 0830) while others consist of a mixture of surface covers (0166, 0313, and 0599). The standard deviation of the albedo of the landscape units consisting of a single surface cover is due to variations in that surface cover from sample to sample. The standard deviation of the albedo of the mixed landscape units is due to variations in the surfaces plus variations in the relative abundance of each surface type from sample to sample.

L349-350, should be rewritten.

We are not sure what the reviewer wants to change here. Nevertheless we have changed the name of the section.

Now it reads:

MODIS albedo and AWS air temperature

  1. It is suggested to show the long-term variations of albedos for 7 landscapes, and discuss their variation reasons. There are some studies about albedos in the Antarctic, please make comparisons with them.

This has been done in Figure 10. There does not seem to be a clear trend. We have added new paragraphs at the end of the manuscript:

The seasonal evolution of albedo from the point of view of landscape units has also been studied at Taylor Valley (Mc Murdo Dry Valleys, Antarctica) [33]. In this work, the authors classify the landscape in three landscape units (glacier, lake and soil) from true color images collected using a camera carried by a helicopter. They obtain that glacier and lake landscape units have a greater albedo variability than soil. This is due to sediment cover in ice-covered lakes and to roughness and debris over glaciers. They conclude that wind-driven snow and sediment redistribution are the driving factor of the spatiotem-poral evolution of albedo. Compared to our research, we have used more landscape unit types and we have been able to establish how and why some landscape units evolve into others, which is the critical point to understand the albedo evolution.

The role of surface cover in the evolution of glaciers has been noted in the case of cryoconite holes, which play an important role in the decrease of glacier albedo and in the promotion of melt [34].

The long-term evolution of albedo over Antarctica has also been investigated [35].  According to these authors, Deception Island should be included in a region of Antarctica where a slow decline of albedo has been observed in the period 1983 - 2009. We have not observed a clear trend in the relative abundance of landscape units, except in the case of 0599 and 0041 + 0166 in the period 2006 – 2015 (Figure 13), which would cause not a de-crease but an increase of albedo. In this respect, we must emphasize the importance of re-gional conditions that some studies at a global scale tend to miss. 

  1. It is also suggested to make comparisons between the field measurement and satellite albedos.

Field data were taken on overcast days. No satellite data are available.

Figure 10, please make changes and show all lines clearly.

We have tried several combinations but this is the best we could do.

Figure 11, should be rewritten, the meanings of blue dots and blue solid line are not clear.

Figure caption has been rewritten:

Monthly relative abundance of landscape unit 0599 (blue dots) and mean daily incident energy density (red dots) for each month between seasons 2005-2006 and 2020-2021. Each point represents a month of a season. The solid lines follow the mean value for each month and is given as a guide for the eye.

It is suggested to report detailed information (e.g., location of sampling site, time periods, sky conditions, Cloudiness) about the field measurements and satellite data for the albedos in a table.

A new Table (Table 2) has been added.

Table 1. Location of the sampling sites, sampling date, time of data acquisition, and sky conditions during sampling for each of the landscape units. LST = Local Solar Time.

L          Location            Date     Time     Sky

0830     62°58’41.7’’ S; 60°40’42.7’’ W   1/18/2019         12:44 – 15:10    Overcast

0736     62°58’52.0’’ S; 60°40’55.6’’ W   1/14/2019         10:32 – 10:46    Overcast

0599     62°58’46.0’’ S; 60°40’41.6’’ W   1/11/2029         10:56 – 11:26    Partly cloudy

0457     62°59’10.9’’ S; 60°40’39.2’’ W   1/09/2019         11:05 – 12:09    Overcast

0313     62°58’39.0’’ S; 60°40’38.3’’ W   1/11/2019         9:00 – 9:30       Partly cloudy

0166     62°59’08.1’’ S; 60°40’43.7’’ W   1/08/2019         12:15 – 12:30    Overcast

0041     62°59’07.8’’ S; 60°40’42.8’’ W   1/08/2019         12:20 – 14:00    Overcast
